# CONTINUAL LEARNING KNOWLEDGE GRAPH EMBEDDINGS FOR DYNAMIC KNOWLEDGE GRAPHS

## ABSTRACT

Knowledge graphs (KG) have shown great power in representing the facts for numerous downstream applications. Notice that the KGs are usually evolving and growing with the development of the real world, due to the change of old knowledge and the emergence of new knowledge, thus the study of dynamic knowledge graphs attracts a new wave. However, conventional work mainly pays attention to learning new knowledge based on existing knowledge while neglecting new knowledge and old knowledge should contribute to each other. Under this circumstance, they cannot tackle the following two challenges: (C1) transfer the knowledge from the old to the new without retraining the entire KG; (C2) alleviate the catastrophic forgetting of old knowledge with new knowledge. To address these issues, we revisit the embedding paradigm for dynamic knowledge graphs and propose a new method termed **C**ontinual **L**earning **K**nowledge **G**raph **E**mbeddings (**CLKGE**). In this paper, we establish a new framework, allowing new and old knowledge to be gained from each other. Specifically, to tackle the (C1), we leverage continual learning to conduct the knowledge transfer and obtain new knowledge based on the old knowledge graph. In the face of (C2), we utilize the energy-based model, learning an energy manifold for the knowledge representations and aligning new knowledge and old knowledge such that their energy on the manifold is minimized, hence can alleviate catastrophic forgetting with the assistance of new knowledge. On top of this, we propose a theoretical guarantee that our model can converge to the optimal solution for the dynamic knowledge graphs. Moreover, we conduct extensive experimental results demonstrating that CLKGE achieves state-of-the-art performance compared with the embedding baselines.

## 1 INTRODUCTION

In the past decades, knowledge graphs (KGs) have attracted extensive attention (Vrandecic & Krötzsch, 2014), promoting the boosting of a number of downstream applications, *e.g.,*, semantic search (Xiong et al., 2017), recommendation systems (Cao et al., 2019), and dialogue systems (Lukovnikov et al., 2017). Typically, to learn the representation of knowledge, knowledge graph embedding (KGE) is regarded as a promising direction, which can learn the vector embeddings for knowledge in a low-dimensional space.

Up to now, there are emerging numerous previous work, which achieves remarkable success (Bordes et al., 2013; Wang et al., 2014; Trouillon et al., 2016; Sun et al., 2019; Zhang et al., 2019). Notice that they are usually focusing on the static knowledge graph while the knowledge graphs are practically dynamic in the real world, *e.g.,* we will encounter the new entities/relations or the meaning of original entities and relations will also change in different periods or snapshots. Under this circumstance, the conventional KGE methods cannot be applied to the real-world setting and hence how to handle the dynamic knowledge becomes a critical problem to be addressed.

To this end, some researchers (Hamaguchi et al., 2017; Wang et al., 2019b) propose to learn the knowledge from scratch every time or simply fine-tune them when meeting new facts. Unfortunately the former will cause huge space loss and time cost and does not leverage the previously learned knowledge. The latter would make new embeddings to overwrite the learned knowledge from old snapshots and likely disrupt previously learned knowledge since the distribution of original entities and new entities may be different. There are also some efforts (Rusu et al., 2016; Lomonaco &

Maltoni, 2017; Daruna et al., 2021) propose to learn new knowledge based on existing knowledge, unfortunately, they neglect the new and old knowledge should facilitate each other or the embedding updating process is heuristic as well as lacks the theoretical guarantee.

Based on the above consideration, *our interest is to develop a method to promote the updating process of old knowledge and the learning process of new knowledge in a unified framework*, where the old knowledge and the new knowledge can contribute to each other. To be concrete, the main challenges are as follows:

**(C1)** How to transfer the knowledge from old entities to the new entities without retraining the entire knowledge graph;

**(C2)** How to alleviate the catastrophic forgetting of old knowledge with the assistance of new knowledge.

To solve these challenges, we propose a novel method named **C**ontinual **L**earning **K**nowledge **G**raph **E**mbedding (**CLKGE**). Specifically, we introduce continual learning for dynamic knowledge graphs embedding and divide the representation learning process into two procedures as shown in Figure 1: the *knowledge transfer* and the *knowledge retention*. Specifically, in the face of challenge (**C1**), we leverage *continual learning* and conduct the knowledge transfer, which can overcome the distribution divergence between new knowledge and old knowledge, and learn new knowledge based on old knowledge without retraining. To tackle the challenge (**C2**), we first learn an energy-based manifold where the representations of knowledge from the current snapshot have higher

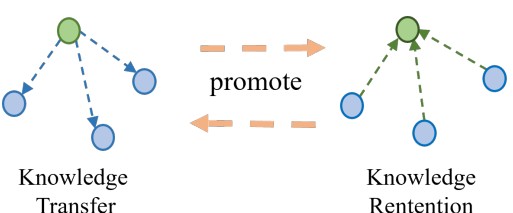

Knowledge Transfer     promote     Knowledge Rentention

Figure 1: Our goal is that knowledge transfer and knowledge rentention can contribute to each other. Green dots denote the old knowledge while blue ones denote new knowledge.

energy, while the counterparts from the previous snapshots have lower energy. Next, the learned energy-based model is used to align new knowledge and old knowledge such that their energy on the manifold is minimized, thus achieving to alleviate the catastrophic forgetting with the assistance of new knowledge. On top of this, we provide the analysis for convergence, which guarantees that our model can achieve the optimal solution in the training. Moreover, experimental results demonstrate the superiority of CLKGE on multi-modal knowledge graph completion tasks.

In a nutshell, the main contributions of this paper are summarized as follows:

- To handle the dynamic knowledge graph in a unified framework, we take the early trial promoting old knowledge and new knowledge to contribute to each other. We propose a new model termed CLKGE and it can conduct updating new knowledge and avoiding catastrophic forgetting of old knowledge in a joint manner.

- Theoretically, we leverage continual learning to achieve knowledge transfer and reformulate the knowledge retain procedure as a aligning problem between old and newknowledge utilizing an energy-based manifold. Moreover, we provide the convergence analysis guaranteeing that CLKGE can converge to the optimal solution in the evolution process.

- Extensive experiments show that CLKGE achieves state-of-the-art performance on four multi-modal knowledge graph completion tasks.

## 2   RELATED WORK

**Conventional Knowledge Graph Completion**. Conventional KG embedding methods usually embed the entities and relations into a low-dimensional space. Specifically, most existing KG embedding models (Dettmers et al., 2017; Schlichtkrull et al., 2018; Guo et al., 2019; Vashishth et al., 2020) focus on static graphs and cannot learn new knowledge on the growing KGs. Notice that some researchers learn to represent an entity by aggregating its existing neighbors in the previous KG snapshot. For instance, MEAN (Hamaguchi et al., 2017) uses a graph convolutional network

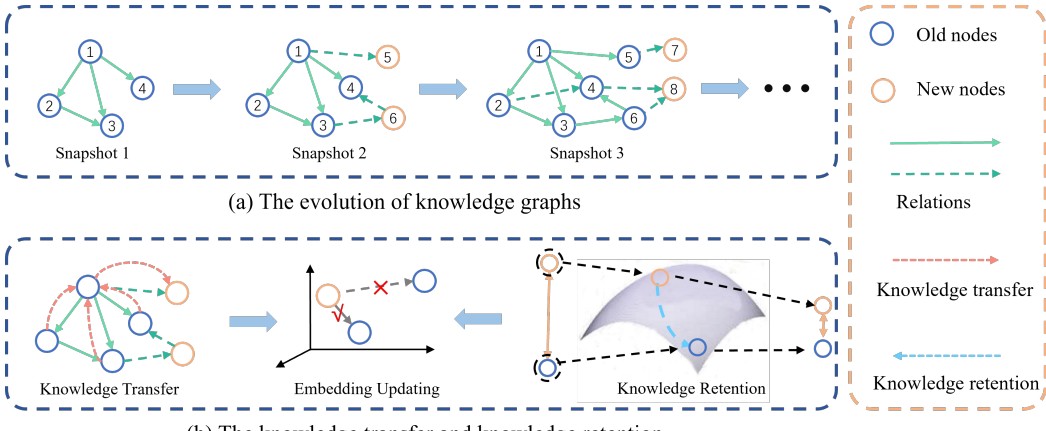

Figure 2: The framework of CLKGE. (a) shows that the knowledge graph is growing among different snapshots. (b) shows that in CLKGE, knowledge transfer, and knowledge retention work together thus conducting the process of embedding updating for the evolution of KGs.

(GCN) (Scarselli et al., 2009) to conduct the neighborhood aggregation. The GCN would aggregate its previously seen neighboring entities to generate an embedding when an unseen entity emerges. Moreover, to attentively aggregate different neighbors, LAN (Wang et al., 2019b) adopts an attention mechanism. Notice the fact that MEAN and LAN rely on the entity neighborhood for embedding learning, one can observe that they cannot handle the new entities that have no neighbors in the previous snapshot.

**Knowledge Graph Embedding for new Entities.** For the new entities, pu-TransE (Tay et al., 2017) trains several new models jointly to handle this case. Inspired by this, DKGE (Wu et al., 2022) learns contextual embeddings for entities and relations, which can be automatically updated as the KG grows. However, they both need partial re-training on old facts, which will spend much time. Moreover, there are also some subgraph-based models, such as GraIL (Teru et al., 2020), INDIGO (Liu et al., 2021), and TACT (Chen et al., 2021), which can also represent unseen entities using the entity-independent features and subgraph aggregation. Their subgraph-building process is time-consuming, making them only applicable to small KGs. Moreover, these are also some current research on temporal knowledge graphs achieving remarkable success (Xu et al., 2023; Jung et al., 2020). Applied the model to large-scale KGs is a promising direction, inspired by the Bellman-Ford algorithm, NBFNet (Zhu et al., 2021) proposes a fast node pair embedding model and NodePiece (Galkin et al., 2022) uses tokenized anchor nodes and relational paths to represent new entities. However, they only focus on the new entities while neglecting the new relations.

**Dynamic Knowledge Graph Embedding**. In the face of dynamic knowledge graphs, dynamic architecture models (Rusu et al., 2016; Lomonaco & Maltoni, 2017) extend the network to learn new tasks and avoid forgetting acquired knowledge. For instance, regularization-based models (Kirkpatrick et al., 2016; Zenke et al., 2017) capture the importance of model parameters for old tasks while adopting the strategy of limiting the update of important parameters. To learn the new knowledge, Rehearsal-based models (Lopez-Paz & Ranzato, 2017; Wang et al., 2019a) memorize some data from old tasks. Moreover, DiCGRL (Kou et al., 2020) splits node embeddings into different components, which can be regarded as a disentangle-based lifelong graph embedding model. Furthermore, LKGE (Daruna et al., 2021) utilizes class-incremental learning models with TransE for lifelong KG embedding. However, the embedding updating process of these methods is heuristic as well as lacking the guarantee of convergence.

## 3 METHODOLOGY

In this section, we introduce a new embedding method for dynamic knowledge graphs, termed *Continual Learning Knowledge Graph Embeddings* (**CLKGE**). Specifically, noticing the represen-

tations of entities in the dynamic may change between different snapshots, we introduce continual learning and divide the learning process into *knowledge transfer* and *knowledge retention*.

**Dynamic KG**. Given a dynamic KG, its growth process can yield a snapshot sequence, *i.e.*, $\mathcal{G} = \{\mathcal{S}_1, \mathcal{S}_2, \cdots, \mathcal{S}_t\}$. Specifically, each snapshot $\mathcal{S}_i$ can be represented as a triplet $(\mathcal{T}_i, \mathcal{E}_i, \mathcal{R}_i)$, where $\mathcal{T}_i, \mathcal{E}_i$ and $\mathcal{R}_i$ denote the fact, entity and relation sets, respectively. Moreover, each fact is denoted in the form of $(h, r, t) \in \mathcal{T}_i$, where $h, t \in \mathcal{E}_i$ are the subject and object entities, respectively, and $r \in \mathcal{R}_i$ denotes the relation. In this process, some old entities, relations, and triples would be removed in dynamic KG or some new counterparts would be added in dynamic KG.

**Continual KG embedding**. Notice that the growing KG will yield a snapshot sequence $\mathcal{G} = \{\mathcal{S}_1, \mathcal{S}_2, \cdots, \mathcal{S}_t\}$ continually, a continual KG embedding model aims to learn the knowledge in these sequences. Specifically, suppose we have a model trained in the previous period $\mathcal{S}_{i-1}$, when the data in the next sequence are fed in, the KG embedding model should update to fit these new facts and learn embeddings for the new entities $\mathcal{E}_{\Delta_i}$ and new relations $\mathcal{R}_{\Delta_i}$ based on the old entities and relations. Distinctions and relevances between continual Learning and dynamic KGs can refer to A.2

### 3.1 KNOWLEDGE TRANSFER FROM THE OLD TO NEW

Given a dynamic KG, we train the entities and relations in the initial snapshot, and in the next snapshot, we may meet new entities, relations, and facts. To avoid the repetitive training of the KG, promoting knowledge transfer from old knowledge to new knowledge is a promising direction. Specifically, it involves two steps: 1) we form the embeddings of old knowledge based on the previous snapshots thus updating the knowledge representations, 2) we learn the new knowledge from the obtained old entities thus achieving the knowledge transfer from old to new.

**Embedding Transfer for Old Knowledge.** Noticing that the representations of knowledge may change in different snapshots, we first learn the representations of old knowledge. Researchers have proposed a number of KGE approaches (*e.g.,* TransE (Bordes et al., 2013), ComplEx (Trouillon et al., 2016)), which can be utilized to exploit the associations between entities and relations, *i.e.*, $t = f_t(h, r)$, where $h, r, t$ denote the embeddings of the subject entity, relation and object entity, respectively. Based on this, we can deduce two transition functions for entity and relation embeddings. The subject entity of $(h, r, t)$ can be represented by $h = f_h(r, t)$, and the relation embedding is $r = f_r(h, t)$. To reduce the reliance on learned data, we use $e_{i-1}$ and $r_{i-1}$ as the approximate average embeddings of $e$ and $r$ in the $i$-1-th snapshot, respectively:

$$e_i = \frac{CL(e_{i-1}) + g\left(\sum_{(e,r,t)\in\mathcal{N}_i(e)} f_h(r_i, t_i) + \sum_{(h,r,e)\in\mathcal{N}_i(e)} f_t(h_i, r_i)\right)}{\sum_{j=1}^{i-1}|\mathcal{N}_j(e)| + |\mathcal{N}_i(e)|}, \quad (1)$$

$$r_i = \frac{CL(r_{i-1}) + g\left(\sum_{(h,r,t)\in\mathcal{N}_i(r)} f_r(h_i, t_i)\right)}{\sum_{j=1}^{i-1}|\mathcal{N}_j(r)| + |\mathcal{N}_i(r)|}. \quad (2)$$

where $CL(e_{i-1}) = \sum_{j=1}^{i-1}|\mathcal{N}_j(e)|\, e_{i-1}$, $CL(r_{i-1}) = \sum_{j=1}^{i-1}|\mathcal{N}_j(r)|\, r_{i-1}$ denotes the representations of $e$ and $r$ in the $i-1$ snapshot in the continual learning, respectively. $\mathcal{N}_j(x) \subseteq \mathcal{D}_j$ is the set of facts containing $x$. Noticing that it may meet the unseen entities or relations during the updating process, here $g$ aims to alleviate the distribution gap between original embeddings and new embeddings, which can be modeled by a network such as an MLP (more details refer to Appendix C).

**Embedding Transfer for New knowledge.** During the evolution of a dynamic KG, there are abundant unseen entities and some unseen relations emerge with the new facts. Notice that these unseen ones are not included in any learned snapshots, so only inheriting the learned parameters cannot transfer the acquired knowledge to their embeddings. To avoid learning from scratch, here we learn the representation of new knowledge by knowledge transfer from the old knowledge. Specifically, we initialize the embeddings of each unseen entity or relation by aggregating its facts:

$$e_i = \frac{1}{|\mathcal{N}_i(e)|} g\left(\sum_{(e,r,t)\in\mathcal{N}_i(e)} f_h(r_{i-1}, t_{i-1}) + \sum_{(h,r,e)\in\mathcal{N}_i(e)} f_t(h_{i-1}, r_{i-1})\right) \quad (3)$$

$$\boldsymbol{r}_i = \frac{1}{|\mathcal{N}_i(r)|} g \left( \sum_{(h,r,t) \in \mathcal{N}_i(r)} f_{\boldsymbol{r}} \left( \boldsymbol{h}_{i-1}, \boldsymbol{t}_{i-1} \right) \right) \tag{4}$$

where $\mathcal{N}_i(e) \subseteq \mathcal{D}_i$ is the set of facts containing $e$. $\mathcal{N}_i(r) \subseteq \mathcal{D}_i$ is the set of facts containing $r$. For the new entities that do not have common facts involving existing entities, we randomly initialize their embeddings, and this strategy is also suit for new entities.

For each snapshot, to learn the knowledge from the new data and update the learned parameters, we leverage the following loss function to train the embedding model:

$$\mathcal{L}_{\text{transfer}} = \sum_{(h,r,t) \in \mathcal{T}_i \cup \mathcal{T}_i^-} \log \left( 1 + \exp \left( -Y_{\text{fact}} f(h,r,t) \right) \right) \tag{5}$$

where $Y_{\text{fact}} \in \{-1, 1\}$ denotes the label of the triple $(h, r, t)$, and $f(h, r, t)$ is the score function (Bordes et al., 2013; Trouillon et al., 2016). Here suppose $\mathcal{T}_i$ denotes the set of observed triples, then let $\mathcal{T}_i^- = \mathcal{E} \times \mathcal{R} \times \mathcal{E} - \mathcal{T}_i$ be the set of unobserved triples. In training, we adopt the negative sampling strategies (*e.g.*, uniform sampling or Bernoulli sampling (Wang et al., 2014)).

## 3.2 KNOWLEDGE RETENTION FROM THE NEW TO OLD

In the process above, we have obtained the representation of entities and relations by knowledge transfer from old knowledge to new knowledge. *However, one can observe that learning new snapshots is likely to overwrite the learned knowledge from old snapshots thus causing the catastrophic forgetting of old knowledge.* Here a critical issue comes as follows: given an entity, how can we leverage the association between new knowledge and old knowledge thus alleviating the catastrophic forgetting?

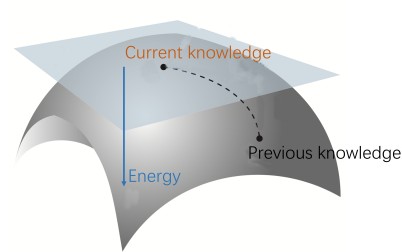

**Energy-based Model (EBM).** As shown in (LeCun et al., 2006), EBMs are a type of maximum likelihood estimation models that can assign low energies to observed data-label pairs and high energies otherwise Du & Mordatch (2019), hence effective in controlling the representational shift that affects incremental models (more details can refer to A.1). Inspired by it, denote $p_{previous}$ is the distribution for previous snapshots and the $p_{\text{current}}$ is the distribution for the current snapshot. In this way, we learn an energy manifold using two ingredients: (i) knowledge representations for entities or relations obtained in the previous snapshots: $\boldsymbol{z}_{previous} \sim p_{previous}$, and (ii) knowledge representations entities or relations in the model trained in the current snapshot: $\boldsymbol{z}_{current} \sim p_{current}$. An energy-based model $\mathcal{E}_\phi$ is learned to assign low energy values for $\boldsymbol{z}_{previous}$, and high energy values for $\boldsymbol{z}_{current}$. In this way, during inference, $\boldsymbol{z}_{current}$ will have higher energy values in the energy manifold. We align $\boldsymbol{z}_{current}$ to $\boldsymbol{z}_{previous}$ by new and old knowledge such that their energy on the manifold is minimized.

Figure 3: An illustration of energy based model. We everage the earned energy-based model used to align the representation of new and old knowledge, thus alleviating catastrophic forgetting with the assistance of new knowledge

In this sense, for the representation $\boldsymbol{z} \in \mathbb{R}^D$ of the knowledge in CLKGE, we learn an energy function $E_\phi(\boldsymbol{z}) : \mathbb{R}^D \to \mathbb{R}$ to map it to a scalar energy value. Specifically, The energy function $E_\phi$ is realized by a multi-layer perceptron with a single neuron in the output layer, which quantifies the energy of the input sample. An EBM is defined as Gibbs distribution $p_\phi(\boldsymbol{z})$ over $E_\phi(\boldsymbol{z})$:

$$p_\psi(\mathbf{z}) = \frac{\exp \left( -E_\psi(\mathbf{z}) \right)}{\int_{\mathbf{z}} \exp \left( -E_\psi(\mathbf{z}) \right) d\mathbf{z}} \tag{6}$$

where $\int_{\boldsymbol{z}} exp(-E_\phi(\boldsymbol{z})) dz$ is an intractable partition function. One can notice that $p_{\text{previous}}$ represents the distribution of latent representations obtained from the model trained on the previous task at any given time. Directly sampling from $p_\phi(\boldsymbol{x})$ is infeasible due to the presence of the normalization constant in Equation (6). Instead, an approximation of the samples is generated through the use of Langevin dynamics (Neal et al., 2011; Welling & Teh, 2011), a well-known MCMC algorithm.

$$\mathbf{z}_{i+1} = \mathbf{z}_i - \frac{\lambda}{2} \partial_{\mathbf{z}} E_\psi(\mathbf{z}) + \sqrt{\lambda} \omega_i, \omega_i \sim \mathcal{N}(0, \mathbf{I}) \tag{7}$$

where $\lambda$ is the step size and $\omega$ captures data uncertainty. Eq.(7) yields a Markov chain that stabilizes to a stationary distribution within a few iterations, starting from an initial $\boldsymbol{z}_i$

**Knowledge Retention Function.** In this way, we have established the knowledge retention module based on the energy-based model. In this way, we can conduct the knowledge updating and alleviate catastrophic forgetting by the following formulation:

$$\mathcal{L}_{retention} = \mathbb{E}_{\mathbf{z} \sim p_{\text{previous}}} \left[ -E_\psi(\mathbf{z}) \right] + \mathbb{E}_{\mathbf{z}_{current} \sim p_\phi} \left[ E_\psi(\mathbf{z}_{current}) \right] \tag{8}$$

In this way, we can mitigate catastrophic forgetting by optimizing this energy function to achieve representational alignment of new knowledge with old knowledge.

**Overall Loss Function.** To promote the process of knowledge transfer and knowledge retention in a unified manner, the overall continual learning objective $\mathcal{L}$ is defined as follows:

$$\mathcal{L} = \beta_1 \mathcal{L}_{\text{transfer}} + \beta_2 \mathcal{L}_{\text{retention}} \tag{9}$$

where $\beta_1$ and $\beta_2$ are hyperparameters for balancing the objectives.

## 4 THEORETICAL ANALYSIS

**Convergence Statement.** We have illustrated the framework of CLKGE with knowledge transfer and knowledge retention processes, here we study the convergence of the algorithm (Garrigos et al., 2023; Lin et al., 2022; Monjezi & Nobakhtian, 2022). To be simplistic, we utilize the gradient descent algorithm to demonstrate it. For ease of exposition, consider the scenario with a sequence of two datasets $\mathcal{D}_{j-1}$ in snapshot $j$-1 and $\mathcal{D}_j$ in snapshot $j$ forming the old embeddings and new embeddings, respectively. Let $\mathcal{F}(w) = \mathcal{L}(w, \mathcal{D}_{j-1}) + \mathcal{L}(w, \mathcal{D}_j)$, $g_1(w) = \Delta_w \mathcal{L}(w, \mathcal{D}_{j-1})$ and $g_2(w) = \Delta_w \mathcal{L}(w, \mathcal{D}_j)$.

Notice that the entities and relations have been trained in the snapshot $j$-1, to avoid the retraining, we consider the updating process of parameters in $\mathcal{D}_j$ based on snapshot $j$-1. We take the following updating formulation for parameters: $\boldsymbol{w}_{k+1} = \boldsymbol{w}_k - \alpha \boldsymbol{g}_2(\boldsymbol{w}_k)$ and $w_0$ is the parameters in snapshot $j$-1. In the updating process, we set $\langle \nabla \mathcal{L}_j(\boldsymbol{w}^j), \nabla \mathcal{L}_{j-1}(\boldsymbol{w}^{j-1}) \rangle \geq \epsilon_1 \left\| \nabla \mathcal{L}_j(\boldsymbol{w}^j) \right\|_2 \left\| \nabla \mathcal{L}_{j-1}(\boldsymbol{w}^{j-1}) \right\|_2$. In this way, we have the following theorem:

**Theorem 1** *Suppose that the loss $\mathcal{L}$ is B-Lipschitz and $\frac{H}{2}$-smooth. Let $\alpha < \min\{\frac{1}{H}, \frac{\gamma \|g_1(w_o)\|}{HBK}\}$ and $\epsilon_1 \geqslant \frac{(2+\gamma^2)\|g_1(w_0)\|}{4\|g_2(w_0)\|}$ for some $\gamma \in (0, 1)$. We have the following results:*

*(1) If $\mathcal{L}$ is convex, we can make the parameters for new snapshot of KG converges to the optimal model $w^* = \arg \min \mathcal{F}(w)$;*

*(2) If $\mathcal{L}$ is nonconvex, we can make the parameters for new snapshot of KG converges to the first order stationary point, i.e.,*

$$\min_k \|\nabla \mathcal{F}(w_k)\|^2 < \frac{2}{\alpha K} \left[ \mathcal{F}(w_0) - \mathcal{F}(w^*) \right] + \frac{4+\gamma^2}{2} \|g_1(w_0)\|^2. \tag{10}$$

Theorem 1 indicates that updating our model will guide to the convergence to the minimizer of the joint objective function $\mathcal{F}(w)$ in the convex case, and the convergence to the first-order stationary point in the nonconvex case when snapshot $i$ and $i-1$ satisfy $\epsilon_1 \geqslant \frac{(2+\gamma^2)\|g_1(w_0)\|}{4\|g_2(w_0)\|}$. That is to say, our model can not only result in a good model for snapshot $i$ but can also be beneficial for the joint learning of snapshot $i-1$ and $i$. Note that since $w_0$ is the learned model of snapshot $i-1$, in general, we have $\|g_1(w_0)\| < \|g_2(w_0)\|$. Proof of Theorem 1 can be found in Appendix B.

**The Connection to Other Models.** Here we take some discussion between CLKGE and other models. First, we design a new knowledge transfer module, which can tackle the heterogeneity of new knowledge and old knowledge while previous work does not take it into consideration. Moreover, we propose a novel strategy for alleviating catastrophic forgetting in an adaptive way and we equip this process with the physical explanation in the real world. Compared with other methods, we take the early trial and provide the theoretical guarantee that our model can converge the optimal solution during training, which is the first to demonstrate the effectiveness of methods in the theory of convergence. Furthermore, we believe that utilizing the physical process to model the embedding updating strategy may be a promising direction.

Table 1: Result comparison for ENTITY and RELATION on the union of the test sets in all snapshots. CLKGE-X (X=S,R,F) denotes different versions of CLKGE.

| Model | ENTITY | | | | RELATION | | | |
|---|---|---|---|---|---|---|---|---|
| | MRR | H@1 | H@3 | H@10 | MRR | H@1 | H@3 | H@10 |
| MEAN | $.117_{\pm.005}$ | $.068_{\pm.003}$ | $.123_{\pm.006}$ | $.212_{\pm.007}$ | $.039_{\pm.004}$ | $.024_{\pm.005}$ | $.040_{\pm.005}$ | $.067_{\pm.008}$ |
| LAN | $.141_{\pm.004}$ | $.082_{\pm.004}$ | $.149_{\pm.003}$ | $.256_{\pm.005}$ | $.052_{\pm.003}$ | $.033_{\pm.003}$ | $.052_{\pm.004}$ | $.092_{\pm.008}$ |
| PNN | $.229_{\pm.001}$ | $.130_{\pm.001}$ | $.265_{\pm.001}$ | $.425_{\pm.001}$ | $.167_{\pm.002}$ | $.096_{\pm.001}$ | $.191_{\pm.001}$ | $.305_{\pm.001}$ |
| CWR | $.088_{\pm.002}$ | $.028_{\pm.001}$ | $.114_{\pm.004}$ | $.202_{\pm.007}$ | $.021_{\pm.000}$ | $.010_{\pm.000}$ | $.024_{\pm.000}$ | $.043_{\pm.000}$ |
| SI | $.154_{\pm.003}$ | $.072_{\pm.003}$ | $.179_{\pm.003}$ | $.311_{\pm.004}$ | $.113_{\pm.002}$ | $.055_{\pm.002}$ | $.131_{\pm.002}$ | $.224_{\pm.002}$ |
| EWC | $.229_{\pm.001}$ | $.130_{\pm.001}$ | $.264_{\pm.002}$ | $.423_{\pm.001}$ | $.165_{\pm.005}$ | $.093_{\pm.005}$ | $.190_{\pm.005}$ | $.306_{\pm.006}$ |
| GEM | $.165_{\pm.002}$ | $.085_{\pm.002}$ | $.188_{\pm.002}$ | $.321_{\pm.002}$ | $.093_{\pm.001}$ | $.040_{\pm.002}$ | $.106_{\pm.002}$ | $.196_{\pm.002}$ |
| EMR | $.171_{\pm.002}$ | $.090_{\pm.001}$ | $.195_{\pm.002}$ | $.330_{\pm.003}$ | $.111_{\pm.002}$ | $.052_{\pm.002}$ | $.126_{\pm.003}$ | $.225_{\pm.004}$ |
| DiCGRL | $.107_{\pm.009}$ | $.057_{\pm.009}$ | $.110_{\pm.008}$ | $.211_{\pm.009}$ | $.133_{\pm.007}$ | $.079_{\pm.005}$ | $.147_{\pm.009}$ | $.241_{\pm.012}$ |
| LKGE | $\underline{.234}_{\pm.001}$ | $\underline{.136}_{\pm.001}$ | $\underline{.269}_{\pm.002}$ | $\underline{.425}_{\pm.003}$ | $\underline{.192}_{\pm.000}$ | $\underline{.106}_{\pm.001}$ | $\underline{.219}_{\pm.001}$ | $\underline{.366}_{\pm.002}$ |
| **CLKGE** | $\mathbf{.248}_{\pm.001}$ | $\mathbf{.144}_{\pm.002}$ | $\mathbf{.278}_{\pm.001}$ | $\mathbf{.436}_{\pm.002}$ | $\mathbf{.203}_{\pm.001}$ | $\mathbf{.115}_{\pm.002}$ | $\mathbf{.226}_{\pm.002}$ | $\mathbf{.379}_{\pm.001}$ |
| CLKGE-S | $.084_{\pm.001}$ | $.028_{\pm.000}$ | $.107_{\pm.002}$ | $.193_{\pm.003}$ | $.021_{\pm.000}$ | $.010_{\pm.000}$ | $.023_{\pm.000}$ | $.043_{\pm.001}$ |
| CLKGE-R | $.236_{\pm.001}$ | $.137_{\pm.001}$ | $.274_{\pm.001}$ | $.433_{\pm.001}$ | $.219_{\pm.001}$ | $.128_{\pm.001}$ | $.250_{\pm.001}$ | $403_{\pm.002}$ |
| CLKGE-F | $165_{\pm.002}$ | $.085_{\pm.002}$ | $.188_{\pm.003}$ | $.321_{\pm.003}$ | $.093_{\pm.003}$ | $.039_{\pm.002}$ | $.106_{\pm.003}$ | $.195_{\pm.007}$ |

## 5 EXPERIMENTS

### 5.1 EXPERIMENT SETTINGS

**Datasets.** As shown in Table 7 in Appendix D, we use four datasets based on FB15K-237 (Toutanova & Chen, 2015), which are entity-centric, relation-centric, fact-centric, and hybrid, and are denoted by ENTITY, RELATION, FACT, and HYBRID, respectively (Cui et al., 2022). Moreover, we also use the dataset WN18RR-5-LS (Shang et al., 2019). In this way, we can simulate a variety of the KG evolution.

**Baselines**. We compare our model with 13 competitors, including (i) three baseline models: snapshot only, re-training, and fine-tuning; (ii) two inductive models: MEAN (Hamaguchi et al., 2017), and LAN (Wang et al., 2019b); (iii) two dynamic architecture models: PNN (Rusu et al., 2016), and CWR (Lomonaco & Maltoni, 2017); (iv) two regularizationbased models: SI (Zenke et al., 2017), and EWC (Kirkpatrick et al., 2016); (v) four rehearsal-based models: GEM (Lopez-Paz & Ranzato, 2017), EMR (Wang et al., 2019a), DiCGRL (Kou et al., 2020), and LKGE (Cui et al., 2023).

**Evaluation Metrics**. Prediction is a typical task in the field of knowledge graphs. Following the conventional methods, we adopt the following metrics to conduct the evaluation for all models: (1) The metrics measuring accuracy for link prediction: mean reciprocal rank (MRR) and Hits@k ($k = 1, 3, 10$, and we denote as H@k). (2) The metrics measuring knowledge transfer capability: forward transfer (FWT) and backward transfer (BWT). Specifically, $FWT = \frac{1}{n-1}\sum_{i=2}^{n} h_{i-1,i}$ is the influence of learning a task on the performance of future tasks (The higher the value, the better the performance), while $BWT = \frac{1}{n-1}\sum_{i=1}^{n-1}(h_{n,i} - h_{i,i})$ is the influence of learning the previous tasks (The smaller the value, the better the performance).

**Implementation Details.** In the experiments, we leverage grid-search to tune the hyperparameters of the baseline model, searching learning rate in {0.0005,0.0001,0.001,0.01}, batch size in {256, 521, 1024, 2048}, embedding dimension in {100, 200, 300}. For the overall loss, the hyperparameter $\beta_1$ is searched in 0.1 to 0.9 and $\beta_2 = 1 - \alpha_1$. For all competitors, we use Adam optimizer and set the patience of early stopping to 5. All experiments are conducted with a single NVIDIA RTX 3090 GPU.

### 5.2 EXPERIMENTAL RESULTS

**Link Prediction.** The experimental results are shown in Table 1 and Table 2. To exclude the effect of random seeds, we conduct link prediction tasks with 5-seed experiments on each dataset. First of all, one can observe that our model can superior other models in four datasets significantly. The reason lies in that CLKGE can promote knowledge transfer and knowledge retention jointly, which can learn new knowledge meanwhile tackling catastrophic forgetting. Specifically, the relations in

Table 2: Result comparison for FACT and HYBRID on the union of the test sets in all snapshots. CLKGE-X (X=S,R,F) denotes different versions of CLKGE

| Model | FACT | | | | HYBRID | | | |
|---|---|---|---|---|---|---|---|---|
| | MRR | H@1 | H@3 | H@10 | MRR | H@1 | H@3 | H@10 |
| MEAN | $.084_{\pm.008}$ | $.051_{\pm.005}$ | $.088_{\pm.008}$ | $.146_{\pm.015}$ | $.046_{\pm.004}$ | $.029_{\pm.003}$ | $.049_{\pm.003}$ | $.080_{\pm.004}$ |
| LAN | $.106_{\pm.007}$ | $.056_{\pm.006}$ | $.113_{\pm.007}$ | $.200_{\pm.011}$ | $.059_{\pm.005}$ | $.032_{\pm.005}$ | $.062_{\pm.005}$ | $.113_{\pm.007}$ |
| PNN | $.157_{\pm.000}$ | $.084_{\pm.002}$ | $.188_{\pm.001}$ | $.290_{\pm.001}$ | $.185_{\pm.001}$ | $.101_{\pm.001}$ | $.216_{\pm.001}$ | $.349_{\pm.001}$ |
| CWR | $.083_{\pm.001}$ | $.030_{\pm.002}$ | $.095_{\pm.002}$ | $.192_{\pm.005}$ | $.037_{\pm.001}$ | $.015_{\pm.001}$ | $.044_{\pm.002}$ | $.077_{\pm.002}$ |
| SI | $.172_{\pm.004}$ | $.088_{\pm.003}$ | $.194_{\pm.004}$ | $.343_{\pm.005}$ | $.111_{\pm.004}$ | $.049_{\pm.003}$ | $.126_{\pm.006}$ | $.229_{\pm.006}$ |
| EWC | $.201_{\pm.001}$ | $.113_{\pm.001}$ | $.229_{\pm.001}$ | $.382_{\pm.001}$ | $.186_{\pm.004}$ | $.102_{\pm.003}$ | $.214_{\pm.004}$ | $.350_{\pm.004}$ |
| GEM | $.175_{\pm.004}$ | $.092_{\pm.003}$ | $.196_{\pm.005}$ | $.345_{\pm.007}$ | $.136_{\pm.003}$ | $.070_{\pm.001}$ | $.152_{\pm.004}$ | $.263_{\pm.005}$ |
| EMR | $.171_{\pm.004}$ | $.090_{\pm.003}$ | $.191_{\pm.004}$ | $.337_{\pm.006}$ | $.141_{\pm.002}$ | $.073_{\pm.001}$ | $.157_{\pm.002}$ | $.267_{\pm.003}$ |
| DiCGRL | $.162_{\pm.007}$ | $.084_{\pm.007}$ | $.189_{\pm.008}$ | $.320_{\pm.007}$ | $.149_{\pm.005}$ | $.083_{\pm.004}$ | $.168_{\pm.005}$ | $.277_{\pm.008}$ |
| LKGE | $\underline{.210}_{\pm.002}$ | $\underline{.122}_{\pm.001}$ | $\underline{.238}_{\pm.002}$ | $\underline{.387}_{\pm.002}$ | $\underline{.207}_{\pm.002}$ | $\underline{.121}_{\pm.002}$ | $\underline{.235}_{\pm.002}$ | $\underline{.379}_{\pm.003}$ |
| **CLKGE** | $\mathbf{.223}_{\pm.003}$ | $\mathbf{.138}_{\pm.002}$ | $\mathbf{.245}_{\pm.001}$ | $\mathbf{.398}_{\pm.003}$ | $\mathbf{.220}_{\pm.004}$ | $\mathbf{.134}_{\pm.003}$ | $\mathbf{.242}_{\pm.001}$ | $\mathbf{.389}_{\pm.002}$ |
| CLKGE-S | $.082_{\pm.001}$ | $.030_{\pm.001}$ | $.095_{\pm.002}$ | $.191_{\pm.006}$ | $.036_{\pm.001}$ | $.015_{\pm.001}$ | $.043_{\pm.001}$ | $.077_{\pm.003}$ |
| CLKGE-R | $.206_{\pm.001}$ | $.118_{\pm.001}$ | $.232_{\pm.001}$ | $.385_{\pm.001}$ | $.227_{\pm.001}$ | $.134_{\pm.001}$ | $.260_{\pm.002}$ | $.413_{\pm.001}$ |
| CLKGE-F | $.172_{\pm.003}$ | $.090_{\pm.002}$ | $.193_{\pm.004}$ | $.339_{\pm.005}$ | $.135_{\pm.002}$ | $.069_{\pm.001}$ | $.151_{\pm.003}$ | $.262_{\pm.005}$ |

datasets such as ENTITY and FACT are stable. One can observe that most models perform well while the performance of GEM, EMR, and DiCGRL is limited. the reason may lie in that these models can only learn the specific entities and relations. As for RELATION and HYBRID, relational patterns are constantly changing due to unseen relations. One can observe that most models perform poorly, which implies that the variation of relational patterns is more challenging for continual KG embedding. It is worth noticing that PNN can conduct the knowledge retention process well but the performance on FACT is not well due to the new entities in datasets. In a nutshell, we can see the ability of knowledge transfer and knowledge retention all play important roles in modeling the dynamic knowledge graphs, demonstrating making these two processes contribute to each other is a feasible direction. We also conduct the experiments on WN18RR-5-LS dataset as shown in Table 6 in Appendix D.

**Evolution Ability of Model.** In this part we demonstrate the ability of our model for evolution during the learning process. Specifically, we evaluate the model $Mi$ the version of CLKGE trained for the i-th snapshot (denoted as Si) using the test data from previous snapshots. The MRR results are shown in Figure 4, and one can observe that CLKGE can consistently maintain the learned knowledge during continual learning. Moreover, one can observe that the knowledge updating process can improve the performance of old test data, *e.g.,* we learn the $M5$ in S5 and the MRR of $M5$ on S5 is 0.20 and its performance on S4 is increased to 0.21. It shows that the embedding learned in the new snapshot can effectively improve the performance in the old snapshot, thus demonstrating the effectiveness of knowledge transfer and knowledge retention.

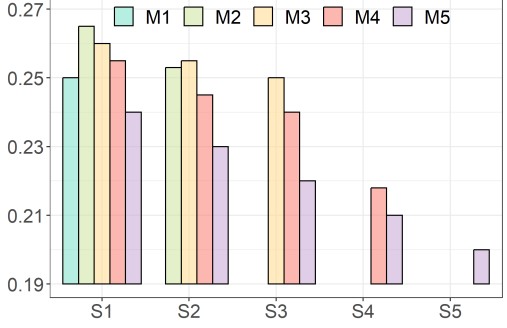

Figure 4: MRR changes for Mi in different snapshots. For instance, M4 is learned in S4, we evaluate it in the test data from previous snapshots hence the bar chart of M4 is empty in S5. The same is true for the other Mi.

**The results for Different Versions.** CLKGE-S denoted the version where we reinitialize and train a model only on the training set $\mathcal{D}_i$. CLKGE-R denotes the version where we reinitialize and train a model for the i-th snapshot on the accumulated training data $\cup_{j=1}^{i} \mathcal{D}_j$. CLKGE-F denotes the version where the model inherits the learned parameters of the model trained on the previous snapshots, and we incrementally train it on $\mathcal{D}_i$. We can see that CLKGE-R can achieve the best performance in all versions. However, the original CLKGE does not differ much from CLKGE-R, which demonstrates the effectiveness of CLKGE without re-training.

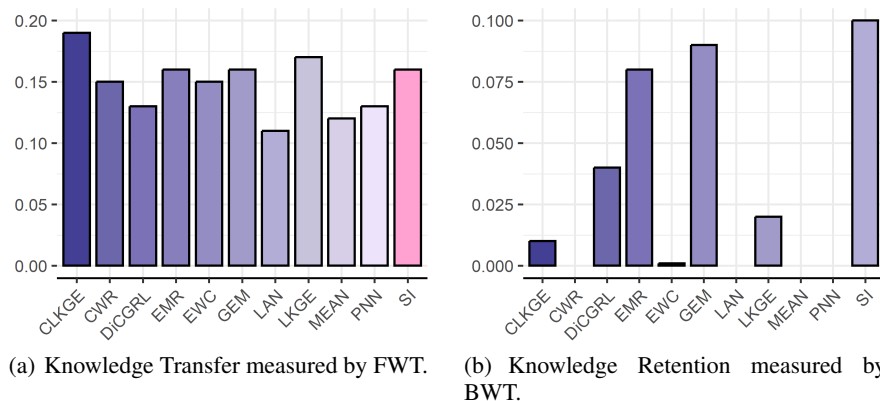

(a) Knowledge Transfer measured by FWT.  (b) Knowledge Retention measured by BWT.

Figure 5: Knowledge transfer and knowledge retention. (a) shows that CLKGE can achieve the best score of FWT. In (b), one can observe that some histograms are empty since these models do not update the model parameters they are not compared.

Table 3: Ablation results of link prediction on the union of the test sets in all snapshots.

| Model | ENTITY | | | | RELATION | | | |
|---|---|---|---|---|---|---|---|---|
| | MRR | H@1 | H@3 | H@10 | MRR | H@1 | H@3 | H@10 |
| w/o transfer | 0.236 | 0.139 | 0.265 | 0.426 | 0.172 | 0.089 | 0.195 | 0.328 |
| w/o retention | 0.161 | 0.083 | 0.178 | 0.309 | 0.137 | 0.072 | 0.146 | 0.285 |

**Knowledge Transfer and Knowledge Retention.** Here we conduct experiments to evaluate the knowledge transfer and retention capability of all models in Figure 5. On one hand, the FWT (The higher the value, the better the performance) of CLKGE is higher than all competitors, which demonstrates the effectiveness of knowledge transfer in CLKGE. On the other hand, notice that PNN, MEAN, and LAN do not update the learned parameters, so their BWT scores are "NA". The poor BWT scores of CWR show the harmful effects of the average operation due to the overwriting of learned knowledge. Overall, CLKGE can achieve good performance of BWT scores as the knowledge retention in CLKGE can well balance the learning of new and the update of old embeddings, which demonstrates the effectiveness of utilizing the energy-based manifold to model the association of true embeddings and learned embeddings.

**Ablation Study.** We conduct an ablation study to validate the effectiveness of each model component. We design two variants termed "w/o transfer" and "w/o retention". Specifically, the "w/o retention" variant is trained on $\mathcal{D}_1$ and performs the embedding transfer on other $\mathcal{D}_i$. As shown in Table 3, on one hand, compared with the original version, the performance of "w/o transfer" is declined. The impact of removing knowledge transfer on the ENTITY dataset is larger than that of ENTITY, probably because the number of relationships is relatively small compared to entities, and learning for unknown relationships is more dependent on knowledge transfer than on entities. On the other hand, deleting knowledge retention can severely influence the performance of our model due to catastrophic forgetting, which demonstrates the importance of knowledge retention.

## 6 CONCLUSION

This paper proposes and studies continual learning for dynamic KGs. Aiming at better knowledge transfer and retention, we propose a continual learning KG embedding model termed **CLKGE**, which can enable old and new knowledge to be gained from each other. Specifically, on one hand, CLKGE introduces continual learning and thus can learn new knowledge based on the old knowledge without retraining the entire knowledge graph. On the other hand, CLKGE leverages the earned energy-based model used to align the representation of new and old knowledge, thus alleviating catastrophic forgetting with the assistance of new knowledge. Moreover, we provide the theoretical guarantee that our model can converge the optimal solution during training. The experimental results on four datasets show better link prediction accuracy, knowledge transfer capability, and learning efficiency of our model. In future work, we plan to investigate continual learning for dynamic KGs for more complex occasions such as temporal settings.

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

# A  DETAILS FOR CLKGE

## A.1  DETAILS FOR EBMS

As shown in (LeCun et al., 2006), theoretical basis for energy assignments in EBMs are as follows:

- Stability and Flexibility: The primary goal of EBMs in a continual learning context is to balance stability with flexibility. Lower energy states are typically associated with more stable, well-learned representations (from previous snapshots), as they are less likely to change. This concept is supported by the stability-plasticity dilemma in neural networks, where stability refers to the retention of old knowledge, and plasticity refers to the ability to adapt to new information. Assigning lower energy to representations from previous snapshots ensures that these stable representations are preserved, mitigating the risk of catastrophic forgetting.

- Prioritizing New Information: Conversely, higher energy states for current snapshot representations imply a more flexible, adaptive state that is still being integrated and solidified within the model's knowledge base. This aligns with the idea that new information requires more attention and resources for effective learning and integration.

- Energy-Based Regularization: The differential energy levels act as a form of regularization, ensuring that the model does not become too rigid (overfitting to old information) or too flexible (overfitting to new information). This concept is akin to the idea of elastic weight consolidation (EWC) in neural networks, which also aims to balance the stability and plasticity through a regularization term.

In our paper, we first learn an energy-based manifold where the representations of knowledge from the current snapshot have higher energy, while the counterparts from the previous snapshots have lower energy. Next, the learned energy-based model is used to align new knowledge and old knowledge such that their energy on the manifold is minimized, thus alleviating the catastrophic forgetting with the assistance of new knowledge as follows:

- Advantages of EBM over Mathematical Optimization: Energy-Based Models (EBMs): as highlighted by (LeCun et al., 2006), EBMs provide a dynamic and adaptable framework compared to traditional mathematical optimization methods, which are often rigid in dynamic environments. EBMs' ability to create energy landscapes for different data states allows for more nuanced control over the learning process. In the context of CLKGE, this adaptability is crucial for handling the evolving nature of knowledge graphs, where new relationships and entities continuously emerge. EBMs can effectively distinguish between and manage these varying data states, which is a significant advantage over traditional optimization methods that may struggle with the dynamic nature of knowledge graphs. The work of (Du & Mordatch, 2019) further supports the efficacy of EBMs in such dynamic settings, emphasizing their flexibility in handling representational shifts and evolving data distributions.

- Mitigating Catastrophic Forgetting with EBM: In CLKGE, the use of EBMs is pivotal in addressing catastrophic forgetting—a critical challenge in continual learning settings. As (LeCun et al., 2006) describe, EBMs maintain lower energy states for previous tasks, ensuring their stability and resistance to disruption by new data. This property is instrumental in CLKGE, where new snapshots of knowledge graphs may otherwise overwrite previously learned knowledge. The energy manifold created by EBMs in CLKGE actively preserves the influence of prior knowledge, allowing the model to seamlessly integrate new information without losing important historical data. This aligning of energy levels is crucial for maintaining the integrity and continuity of the knowledge graph over time, as emphasized by (Du & Mordatch, 2019).

- EBM's Role in Effective Knowledge Transfer: In the CLKGE framework, EBMs play a crucial role in facilitating effective knowledge transfer, particularly in the context of dynamic learning environments like knowledge graphs. By minimizing the energy difference between new and old knowledge, EBMs ensure that latent representations of both remain closely integrated. This approach is essential for maintaining the coherence and accuracy of the knowledge graph as it evolves, allowing the model to adapt to new information while

retaining crucial aspects of previous knowledge. In CLKGE, this energy alignment strategy ensures that knowledge transfer is not just about adding new information, but also about preserving and incorporating existing knowledge. The work of (LeCun et al., 2006) and (Du & Mordatch, 2019) underscores the importance of such an energy-based approach in maintaining balance and coherence in continually evolving systems.

## A.2 Distinction Between Continual Learning and Dynamic KGs

Dynamic graphs and continual learning are interconnected concepts in the field of artificial intelligence, particularly in knowledge graph embedding. Dynamic graphs represent evolving systems where entities (nodes) and their relationships (edges) change over time. Continual learning, on the other hand, focuses on a model's ability to continuously adapt and learn from new data while retaining previously acquired knowledge. In the context of knowledge graphs, this relationship becomes crucial as the graph evolves, requiring the model to adapt without losing historical information. While both dynamic graphs and continual learning address changes over time, they tackle different aspects:

- Dynamic Graphs: Concentrate on the structural evolution of the data. In knowledge graphs, this means accommodating new entities, facts, and relationships, as well as updating or removing outdated information.

- Continual Learning: Focuses on the learning process itself. In the context of knowledge graphs, it's about updating the model's understanding and representation of the graph while preventing the loss of previously learned knowledge, known as catastrophic forgetting.

In our CLKGE framework, we effectively integrate these concepts to address the unique challenges posed by dynamic knowledge graphs:

- Knowledge Transfer (Addressing Challenge C1)**: We leverage continual learning to facilitate the transfer of knowledge from old to new in the dynamic KG. This is done by updating the embeddings of old knowledge based on previous graph snapshots and using these updated embeddings to inform and shape the understanding of new knowledge. This approach negates the need for retraining the model with each new graph snapshot, thereby efficiently managing the graph's evolution.

- Knowledge Retention (Addressing Challenge C2)**: To tackle catastrophic forgetting, we employ an energy-based model (EBM) within the continual learning framework. This EBM aligns the knowledge representations of new and old information on an energy manifold, minimizing their energy differences. By doing so, we ensure that new knowledge embeddings are aligned with, and do not overwrite, the old ones. This method is crucial in retaining the integrity of historical knowledge in the face of new data.

- Unified Framework and Loss Function**: The CLKGE model unifies the processes of knowledge transfer and retention. It uses a combined loss function, balanced by hyperparameters, to ensure that both objectives are met. This unified approach not only maintains the dynamic nature of the KG but also ensures that the learning process is cumulative and coherent over time.

In summary, our CLKGE framework demonstrates a sophisticated integration of continual learning within the realm of dynamic knowledge graphs, addressing both the transfer of knowledge from old to new and the mitigation of catastrophic forgetting. This approach represents a significant advancement in the field, ensuring that knowledge graphs remain accurate, up-to-date, and comprehensive as they evolve.

## B    Proof for Theorem 1

First, we can easily prove the cross-entropy loss is $L_1$-Smooth. Then we prove the $\mathcal{L}_{\text{rentention}}$ is $L_2$-Smooth. To simplify, we set $k = 1$. Then we have $\Delta_y x(y) = \frac{1}{1-y} + \frac{1}{2}\frac{1-2y}{y-y^2}$. Notice that we set $0 < \|y\| < 1$, hence we can easily prove that $|\Delta_{y_1} - \Delta_{y_2}| \leq L_1\|y_1 - y_2\|$. In this way, we can have

that the loss function in Eq.(9) is $H/2$-smooth loss function (we can let $H/2 = \alpha L_1 + \beta L_2$), and it can be easily shown that $\mathcal{F}$ is $H$-smooth.

**(1)** For any $k \in [0, K]$, we can have

$$
\begin{aligned}
\mathcal{F}(\boldsymbol{w}_{k+1}) &\leq \mathcal{F}(\boldsymbol{w}_k) + \nabla \mathcal{F}(\boldsymbol{w}_k)^T (\boldsymbol{w}_{k+1} - \boldsymbol{w}_k) + \frac{H}{2} \|\boldsymbol{w}_{k+1} - \boldsymbol{w}_k\|^2 \\
&= \mathcal{F}(\boldsymbol{w}_k) + (\boldsymbol{g}_1(\boldsymbol{w}_k) + \boldsymbol{g}_2(\boldsymbol{w}_k))^T (-\alpha \boldsymbol{g}_2(\boldsymbol{w}_k)) + \frac{\alpha^2 H}{2} \|\boldsymbol{g}_2(\boldsymbol{w}_k)\|^2 \\
&= \mathcal{F}(\boldsymbol{w}_k) - \left[ \alpha - \frac{\alpha^2 H}{2} \right] \|\boldsymbol{g}_2(\boldsymbol{w}_k)\|^2 - \alpha \langle \boldsymbol{g}_1(\boldsymbol{w}_k), \boldsymbol{g}_2(\boldsymbol{w}_k) \rangle
\end{aligned}
\tag{11}
$$

For the term $\langle g_1(w_k), g_2(w_k) \rangle$, it follows that

$$
\begin{aligned}
&\langle \boldsymbol{g}_1(\boldsymbol{w}_k), \boldsymbol{g}_2(\boldsymbol{w}_k) \rangle \\
&= \langle \boldsymbol{g}_1(\boldsymbol{w}_k) - \boldsymbol{g}_1(\boldsymbol{w}_0) + \boldsymbol{g}_1(\boldsymbol{w}_0), \boldsymbol{g}_2(\boldsymbol{w}_k) \rangle \\
&= \langle \boldsymbol{g}_1(\boldsymbol{w}_k) - \boldsymbol{g}_1(\boldsymbol{w}_0), \boldsymbol{g}_2(\boldsymbol{w}_k) \rangle + \langle \boldsymbol{g}_1(\boldsymbol{w}_0), \boldsymbol{g}_2(\boldsymbol{w}_k) \rangle \\
&= \langle \boldsymbol{g}_1(\boldsymbol{w}_k) - \boldsymbol{g}_1(\boldsymbol{w}_0), \boldsymbol{g}_2(\boldsymbol{w}_k) \rangle + \langle \boldsymbol{g}_1(\boldsymbol{w}_0), \boldsymbol{g}_2(\boldsymbol{w}_k) - \boldsymbol{g}_2(\boldsymbol{w}_0) \rangle + \langle \boldsymbol{g}_1(\boldsymbol{w}_0), \boldsymbol{g}_2(\boldsymbol{w}_0) \rangle
\end{aligned}
\tag{12}
$$

Notice that

$$
\begin{aligned}
&2 \langle \boldsymbol{g}_1(\boldsymbol{w}_k) - \boldsymbol{g}_1(\boldsymbol{w}_0), \boldsymbol{g}_2(\boldsymbol{w}_k) \rangle + \|\boldsymbol{g}_1(\boldsymbol{w}_k) - \boldsymbol{g}_1(\boldsymbol{w}_0)\|^2 + \|\boldsymbol{g}_2(\boldsymbol{w}_k)\|^2 \\
&= \|\boldsymbol{g}_1(\boldsymbol{w}_k) - \boldsymbol{g}_1(\boldsymbol{w}_0) + \boldsymbol{g}_2(\boldsymbol{w}_k)\|^2 \geq 0
\end{aligned}
\tag{13}
$$

We have

$$
\langle \boldsymbol{g}_1(\boldsymbol{w}_k) - \boldsymbol{g}_1(\boldsymbol{w}_0), \boldsymbol{g}_2(\boldsymbol{w}_k) \rangle \geq -\frac{1}{2} \|\boldsymbol{g}_1(\boldsymbol{w}_k) - \boldsymbol{g}_1(\boldsymbol{w}_0)\|^2 - \frac{1}{2} \|\boldsymbol{g}_2(\boldsymbol{w}_k)\|^2
\tag{14}
$$

Following the same line, it can be shown that

$$
\langle \boldsymbol{g}_1(\boldsymbol{w}_0), \boldsymbol{g}_2(\boldsymbol{w}_k) - \boldsymbol{g}_2(\boldsymbol{w}_0) \rangle \geq -\frac{1}{2} \|\boldsymbol{g}_2(\boldsymbol{w}_k) - \boldsymbol{g}_2(\boldsymbol{w}_0)\|^2 - \frac{1}{2} \|\boldsymbol{g}_1(\boldsymbol{w}_0)\|^2
\tag{15}
$$

Combining Eq.(12), Eq.(14) and Eq.(15) gives a lower bound on $g_1(w_k), g_2(w_k)$, *i.e.*,

$$
\begin{aligned}
&\langle \boldsymbol{g}_1(\boldsymbol{w}_k), \boldsymbol{g}_2(\boldsymbol{w}_k) \rangle \\
&\geq -\frac{1}{2} \|\boldsymbol{g}_1(\boldsymbol{w}_k) - \boldsymbol{g}_1(\boldsymbol{w}_0)\|^2 - \frac{1}{2} \|\boldsymbol{g}_2(\boldsymbol{w}_k)\|^2 \\
&\quad -\frac{1}{2} \|\boldsymbol{g}_2(\boldsymbol{w}_k) - \boldsymbol{g}_2(\boldsymbol{w}_0)\|^2 - \frac{1}{2} \|\boldsymbol{g}_1(\boldsymbol{w}_0)\|^2 + \langle \boldsymbol{g}_1(\boldsymbol{w}_0), \boldsymbol{g}_2(\boldsymbol{w}_0) \rangle \\
&\geq -\frac{H^2}{8} \|\boldsymbol{w}_k - \boldsymbol{w}_0\|^2 - \frac{1}{2} \|\boldsymbol{g}_2(\boldsymbol{w}_k)\|^2 \\
&\quad -\frac{H^2}{8} \|\boldsymbol{w}_k - \boldsymbol{w}_0\|^2 - \frac{1}{2} \|\boldsymbol{g}_1(\boldsymbol{w}_0)\|^2 + \langle \boldsymbol{g}_1(\boldsymbol{w}_0), \boldsymbol{g}_2(\boldsymbol{w}_0) \rangle \\
&= -\frac{H^2}{4} \|\boldsymbol{w}_k - \boldsymbol{w}_0\|^2 - \frac{1}{2} \|\boldsymbol{g}_2(\boldsymbol{w}_k)\|^2 - \frac{1}{2} \|\boldsymbol{g}_1(\boldsymbol{w}_0)\|^2 + \langle \boldsymbol{g}_1(\boldsymbol{w}_0), \boldsymbol{g}_2(\boldsymbol{w}_0) \rangle,
\end{aligned}
\tag{16}
$$

where the second inequality is true because of the smoothness of the loss function. Based on the update formulation, it can be seen that

$$
\boldsymbol{w}_k = \boldsymbol{w}_0 - \alpha \sum_{i=0}^{k-1} \boldsymbol{g}_2(\boldsymbol{w}_i)
\tag{17}
$$

Therefore, continuing with Eq.(11), we can have

$$
\begin{aligned}
&\mathcal{F}(\boldsymbol{w}_{k+1}) \\
&\leq \mathcal{F}(\boldsymbol{w}_k) - \left[ \alpha - \frac{\alpha^2 H}{2} \right] \|\boldsymbol{g}_2(\boldsymbol{w}_k)\|^2 - \alpha \langle \boldsymbol{g}_1(\boldsymbol{w}_k), \boldsymbol{g}_2(\boldsymbol{w}_k) \rangle
\end{aligned}
\tag{18}
$$

Then we have:

$$\mathcal{F}\left(\boldsymbol{w}_{k+1}\right)$$

$$\leq \mathcal{F}\left(\boldsymbol{w}_{k}\right) - \left[\alpha - \frac{\alpha^2 H}{2}\right] \left\|\boldsymbol{g}_2\left(\boldsymbol{w}_k\right)\right\|^2 + \frac{\alpha^3 H^2}{4} \left\|\sum_{i=0}^{k-1} \boldsymbol{g}_2\left(\boldsymbol{w}_i\right)\right\|^2 + \frac{\alpha}{2} \left\|\boldsymbol{g}_2\left(\boldsymbol{w}_k\right)\right\|^2$$

$$+ \frac{\alpha}{2} \left\|\boldsymbol{g}_1\left(\boldsymbol{w}_0\right)\right\|^2 - \alpha \left\langle \boldsymbol{g}_1\left(\boldsymbol{w}_0\right), \boldsymbol{g}_2\left(\boldsymbol{w}_0\right)\right\rangle$$

$$= \mathcal{F}\left(\boldsymbol{w}_{k}\right) - \left[\frac{\alpha}{2} - \frac{\alpha^2 H}{2}\right] \left\|\boldsymbol{g}_2\left(\boldsymbol{w}_k\right)\right\|^2 + \frac{\alpha^3 H^2}{4} \left\|\sum_{i=0}^{k-1} \boldsymbol{g}_2\left(\boldsymbol{w}_i\right)\right\|^2 + \frac{\alpha}{2} \left\|\boldsymbol{g}_1\left(\boldsymbol{w}_0\right)\right\|^2 - \alpha \left\langle \boldsymbol{g}_1\left(\boldsymbol{w}_0\right), \boldsymbol{g}_2\left(\boldsymbol{w}_0\right)\right\rangle$$

$$\leq \mathcal{F}\left(\boldsymbol{w}_{k}\right) - \left[\frac{\alpha}{2} - \frac{\alpha^2 H}{2}\right] \left\|\boldsymbol{g}_2\left(\boldsymbol{w}_k\right)\right\|^2 + \frac{\alpha^3 H^2}{4} \left\|\sum_{i=0}^{k-1} \boldsymbol{g}_2\left(\boldsymbol{w}_i\right)\right\|^2 + \frac{\alpha}{2} \left\|\boldsymbol{g}_1\left(\boldsymbol{w}_0\right)\right\|^2$$

$$- \alpha\epsilon_1 \left\|\boldsymbol{g}_1\left(\boldsymbol{w}_0\right)\right\| \left\|\boldsymbol{g}_2\left(\boldsymbol{w}_0\right)\right\|,$$

(19)

where the last inequality is based on $\left\langle \nabla \mathcal{L}_j\left(\boldsymbol{w}^j\right), \nabla \mathcal{L}_{j-1}\left(\boldsymbol{w}^{j-1}\right)\right\rangle \geq \epsilon_1 \left\|\nabla \mathcal{L}_j\left(\boldsymbol{w}^j\right)\right\|_2 \left\|\nabla \mathcal{L}_{j-1}\left(\boldsymbol{w}^{j-1}\right)\right\|_2$. Next, it can be shown that

$$\alpha \leq \frac{\gamma \left\|\boldsymbol{g}_1\left(\boldsymbol{w}_0\right)\right\|}{HBK} \leq \frac{\gamma \left\|\boldsymbol{g}_1\left(\boldsymbol{w}_0\right)\right\|}{H \left\|\sum_{i=0}^{k-1} \boldsymbol{g}_2\left(\boldsymbol{w}_i\right)\right\|}$$

(20)

It then follows that

$$\frac{1}{2} \left\|\boldsymbol{g}_1\left(\boldsymbol{w}_0\right)\right\|^2 + \frac{\alpha^2 H^2}{4} \left\|\sum_{i=0}^{k-1} \boldsymbol{g}_2\left(\boldsymbol{w}_i\right)\right\|^2$$

$$\leq \frac{1}{2} \left\|\boldsymbol{g}_1\left(\boldsymbol{w}_0\right)\right\|^2 + \frac{\gamma^2 \left\|\boldsymbol{g}_1\left(\boldsymbol{w}_0\right)\right\|^2}{4H^2 \left\|\sum_{i=0}^{k-1} \boldsymbol{g}_2\left(\boldsymbol{w}_i\right)\right\|^2} H^2 \left\|\sum_{i=0}^{k-1} \boldsymbol{g}_2\left(\boldsymbol{w}_i\right)\right\|^2$$

(21)

$$= \frac{2 + \gamma^2}{4} \left\|\boldsymbol{g}_1\left(\boldsymbol{w}_0\right)\right\|^2.$$

Therefore, we can obtain that

$$\mathcal{F}\left(\boldsymbol{w}_{k+1}\right) \leq \mathcal{F}\left(\boldsymbol{w}_{k}\right) - \left[\frac{\alpha}{2} - \frac{\alpha^2 H}{2}\right] \left\|\boldsymbol{g}_2\left(\boldsymbol{w}_k\right)\right\|^2 + \frac{\alpha\left(2 + \gamma^2\right)}{4} \left\|\boldsymbol{g}_1\left(\boldsymbol{w}_0\right)\right\|^2 - \alpha\epsilon_1 \left\|\boldsymbol{g}_1\left(\boldsymbol{w}_0\right)\right\| \left\|\boldsymbol{g}_2\left(\boldsymbol{w}_0\right)\right\|$$

$$\leq \mathcal{F}\left(\boldsymbol{w}_{k}\right) - \left[\frac{\alpha}{2} - \frac{\alpha^2 H}{2}\right] \left\|\boldsymbol{g}_2\left(\boldsymbol{w}_k\right)\right\|^2$$

$$< \mathcal{F}\left(\boldsymbol{w}_{k}\right)$$

(22)

where the second inequality is true because $\epsilon_1 \geqslant \frac{\left(2 + \gamma^2\right)\|g_1(w_0)\|}{4\|g_2(w_0)\|}$. This sufficient decrease of the objective function value indicates that the optimal $\mathcal{F}(w*)$ can be obtained eventually for convex loss functions.

**(2)** For a non-convex loss function $\mathcal{L}$, we can have the following as in Eq.(11):

$$
\begin{aligned}
\mathcal{F}\left(w_{k+1}^{r}\right) \leq & \mathcal{F}\left(\boldsymbol{w}_{k}\right)-\left[\alpha-\frac{\alpha^{2} H}{2}\right]\left\|\boldsymbol{g}_{2}\left(\boldsymbol{w}_{k}\right)\right\|^{2}-\alpha\left\langle\boldsymbol{g}_{1}\left(\boldsymbol{w}_{k}\right), \boldsymbol{g}_{2}\left(\boldsymbol{w}_{k}\right)\right\rangle \\
\stackrel{(a)}{=} & \mathcal{F}\left(\boldsymbol{w}_{k}\right)-\left[\alpha-\frac{\alpha^{2} H}{2}\right]\left\|\boldsymbol{g}_{2}\left(\boldsymbol{w}_{k}\right)\right\|^{2}-\frac{\alpha}{2}\left[\left\|\nabla \mathcal{F}\left(\boldsymbol{w}_{k}\right)\right\|^{2}-\left\|\boldsymbol{g}_{1}\left(\boldsymbol{w}_{k}\right)\right\|^{2}-\left\|\boldsymbol{g}_{2}\left(\boldsymbol{w}_{k}\right)\right\|^{2}\right] \\
= & \mathcal{F}\left(\boldsymbol{w}_{k}\right)-\left[\frac{\alpha}{2}-\frac{\alpha^{2} H}{2}\right]\left\|\boldsymbol{g}_{2}\left(\boldsymbol{w}_{k}\right)\right\|^{2}-\frac{\alpha}{2}\left\|\nabla \mathcal{F}\left(\boldsymbol{w}_{k}\right)\right\|^{2}+\frac{\alpha}{2}\left\|\boldsymbol{g}_{1}\left(\boldsymbol{w}_{k}\right)\right\|^{2} \\
= & \mathcal{F}\left(\boldsymbol{w}_{k}\right)-\left[\frac{\alpha}{2}-\frac{\alpha^{2} H}{2}\right]\left\|\boldsymbol{g}_{2}\left(\boldsymbol{w}_{k}\right)\right\|^{2}-\frac{\alpha}{2}\left\|\nabla \mathcal{F}\left(\boldsymbol{w}_{k}\right)\right\|^{2}+\frac{\alpha}{2}\left\|\boldsymbol{g}_{1}\left(\boldsymbol{w}_{k}\right)-\boldsymbol{g}_{1}\left(\boldsymbol{w}_{0}\right)+\boldsymbol{g}_{1}\left(\boldsymbol{w}_{0}\right)\right\|^{2} \\
\leq & \mathcal{F}\left(\boldsymbol{w}_{k}\right)-\left[\frac{\alpha}{2}-\frac{\alpha^{2} H}{2}\right]\left\|\boldsymbol{g}_{2}\left(\boldsymbol{w}_{k}\right)\right\|^{2}-\frac{\alpha}{2}\left\|\nabla \mathcal{F}\left(\boldsymbol{w}_{k}\right)\right\|^{2}+\alpha\left\|\boldsymbol{g}_{1}\left(\boldsymbol{w}_{k}\right)-\boldsymbol{g}_{1}\left(\boldsymbol{w}_{0}\right)\right\|^{2} \\
& +\alpha\left\|\boldsymbol{g}_{1}\left(\boldsymbol{w}_{0}\right)\right\|^{2} \\
\stackrel{(b)}{\leq} & \mathcal{F}\left(\boldsymbol{w}_{k}\right)-\left[\frac{\alpha}{2}-\frac{\alpha^{2} H}{2}\right]\left\|\boldsymbol{g}_{2}\left(\boldsymbol{w}_{k}\right)\right\|^{2}-\frac{\alpha}{2}\left\|\nabla \mathcal{F}\left(\boldsymbol{w}_{k}\right)\right\|^{2}+\frac{H^{2} \alpha^{3}}{4}\left\|\sum_{i=0}^{k-1} \boldsymbol{g}_{2}\left(\boldsymbol{w}_{i}\right)\right\|^{2}+\alpha\left\|\boldsymbol{g}_{1}\left(\boldsymbol{w}_{0}\right)\right\|^{2}
\end{aligned}
$$

$$(23)$$

where (a) is because $\delta \mathcal{F}(w_k) = \boldsymbol{g}_1(\boldsymbol{w}_k) + \boldsymbol{g}_2(\boldsymbol{w}_k)$, and (b) is because of the smoothness of $\mathcal{L}$ and Eq.(17). Therefore,

$$
\begin{aligned}
& \min_{k}\left\|\nabla \mathcal{F}\left(\boldsymbol{w}_{k}\right)\right\|^{2} \\
\leq & \frac{1}{K} \sum_{k=0}^{K-1}\left\|\nabla \mathcal{F}\left(\boldsymbol{w}_{k}\right)\right\|^{2} \\
\leq & \frac{2}{\alpha K} \sum_{k=0}^{K-1}\left[\mathcal{F}\left(\boldsymbol{w}_{k}\right)-\mathcal{F}\left(\boldsymbol{w}_{k+1}\right)+\frac{H^{2} \alpha^{3}}{4}\left\|\sum_{i=0}^{k-1} \boldsymbol{g}_{2}\left(\boldsymbol{w}_{i}\right)\right\|^{2}+\alpha\left\|\boldsymbol{g}_{1}\left(\boldsymbol{w}_{0}\right)\right\|^{2}-\left[\frac{\alpha}{2}-\frac{\alpha^{2} H}{2}\right]\left\|\boldsymbol{g}_{2}\left(\boldsymbol{w}_{k}\right)\right\|^{2}\right] \\
\leq & \frac{2}{\alpha K}\left[\mathcal{F}\left(\boldsymbol{w}_{0}\right)-\mathcal{F}\left(\boldsymbol{w}_{K}\right)\right]+\frac{H^{2} \alpha^{2}}{2(K-1)} \sum_{k=1}^{K-1}\left\|\sum_{i=0}^{k-1} \boldsymbol{g}_{2}\left(\boldsymbol{w}_{i}\right)\right\|^{2}+2\left\|\boldsymbol{g}_{1}\left(\boldsymbol{w}_{0}\right)\right\|^{2}-\frac{1-\alpha H}{K} \sum_{k=0}^{K-1}\left\|\boldsymbol{g}_{2}\left(\boldsymbol{w}_{k}\right)\right\|^{2} \\
\stackrel{(a)}{\leq} & \frac{2}{\alpha K}\left[\mathcal{F}\left(\boldsymbol{w}_{0}\right)-\mathcal{F}\left(\boldsymbol{w}_{K}\right)\right]+\frac{\gamma^{2}}{2}\left\|\boldsymbol{g}_{1}\left(\boldsymbol{w}_{0}\right)\right\|^{2}+2\left\|\boldsymbol{g}_{1}\left(\boldsymbol{w}_{0}\right)\right\|^{2}-\frac{1-\alpha H}{K} \sum_{k=0}^{K-1}\left\|\boldsymbol{g}_{2}\left(\boldsymbol{w}_{k}\right)\right\|^{2} \\
\leq & \frac{2}{\alpha K}\left[\mathcal{F}\left(\boldsymbol{w}_{0}\right)-\mathcal{F}\left(\boldsymbol{w}^{*}\right)\right]+\frac{4+\gamma^{2}}{2}\left\|\boldsymbol{g}_{1}\left(\boldsymbol{w}_{0}\right)\right\|^{2}-\frac{1-\alpha H}{K} \sum_{k=0}^{K-1}\left\|\boldsymbol{g}_{2}\left(\boldsymbol{w}_{k}\right)\right\|^{2} \\
\leq & \frac{2}{\alpha K}\left[\mathcal{F}\left(\boldsymbol{w}_{0}\right)-\mathcal{F}\left(\boldsymbol{w}^{*}\right)\right]+\frac{4+\gamma^{2}}{2}\left\|\boldsymbol{g}_{1}\left(\boldsymbol{w}_{0}\right)\right\|^{2}
\end{aligned}
$$

$$(24)$$

where (a) holds due to $\mathcal{F}(w*) \leq \mathcal{F}(\boldsymbol{w}_K)$ and $\|\sum_{i=0}^{k-1} \boldsymbol{g}_2(\boldsymbol{w}_i)\|^2 \leq \frac{\gamma^2}{\alpha^2 H^2}\|\boldsymbol{g}_1(\boldsymbol{w}_0)\|^2$ based on Eq.(21).

## C DETAILS FOR CLKGE

**Abalation Study for $g$ in Eq.(1).** However, new entities may obey different distributions, hence we need to alleviate the distribution gaps by a function $g$. To achieve this, as stated in (Yang et al., 2022), some networks such as MLP can eliminate the difference in distribution by constructing an implicit space. In this way, for the sake of simplicity, we specify $g$ as an MLP. Furthermore, we conduct the experiments on the ENTITY dataset as shown in Table 4, where CLKGE w/ $g$ denotes CLKGE utilizing the and CLKGE w/o $g$ denotes CLKGE remove $g$. One can observe that CLKGE w/ $g$ can be superior to CLKGE w/o $g$ significantly, which demonstrates the effectiveness of $g$.

**Abalation Study for different KGE methods.** Different KGE methods have somewhat impact on performance. To demonstrate this point, we conduct the experiments on ENTITY datasets utilizing

Table 4: Abalation Study for $g$ in Eq.(1)

| Models | MRR | H@1 | H@3 | H@10 |
|---|---|---|---|---|
| CLKGE w g | $.233_{\pm.002}$ | $.138_{\pm.001}$ | $.245_{\pm.002}$ | $.398_{\pm.002}$ |
| CLKGE w/o g | $.211_{\pm.003}$ | $.120_{\pm.002}$ | $.233_{\pm.002}$ | $.387_{\pm.003}$ |

Table 5: Abalation Study for different KGE methods

| Models | MRR | H@1 | H@3 | H@10 |
|---|---|---|---|---|
| $CLKGE_C$ | $.248_{\pm.001}$ | $.144_{\pm.001}$ | $.278_{\pm.002}$ | $.436_{\pm.002}$ |
| $CLKGE_R$ | $.250_{\pm.002}$ | $.145_{\pm.002}$ | $.279_{\pm.002}$ | $.439_{\pm.003}$ |
| $CLKGE_T$ | $.245_{\pm.003}$ | $.143_{\pm.002}$ | $.274_{\pm.003}$ | $.435_{\pm.003}$ |

the KGE method ComplEx, RotatE, and TransE as $CLKGE_C$, $CLKGE_R$, and $CLKGE_T$, respectively. As shown in the table 5, one can observe that the difference of performance for various KGE method is not big, which also show the robustness of our method.

**Experiments for WN18RR-5-LS dataset.** We conduct experiments on the WN18RR-5-LS dataset. Specifically, WN18RR-5-LS is divided into several snapshots. As shown in the table below, one can notice that CLKGE superiors other methods, and the overall performance shows the effectiveness of components including knowledge transfer and knowledge retention.

# D  EXPERIMENTAL DETAILS

**Learning Efficiency.** In this part, we conduct experiments to compare the training time. We report the total time cost on FACT, which is easier for comparison. Table 8 shows the results. Unsurprisingly, one can observe that re-training is the most time-consuming. By contrast, our model is the most efficient, and it can converge to the optimal embedding in a fast way. The reason is twofold. On one hand, the new knowledge can learn the representations via knowledge transfer by old knowledge effectively. On the other hand, old knowledge can alleviate catastrophic forgetting via knowledge retention with new knowledge significantly. In total, these components work together thus reducing the training time, which demonstrates the effectiveness of our method.

**Ablation Study for Knowledge Retention** To demonstrate the effectiveness of utilizing the energy-based manifold, we compare the performance of two versions. The first is the original version (denoted as $\text{CLKGE}_1$) while the retention loss in the second version (denoted as $\text{CLKGE}_2$) is

$$\mathcal{L}_{\text{retention}} = \sum_{e \in \mathcal{E}_{i-1}} \omega(e) \left\| \mathbf{e}_i - \mathbf{e}_{i-1} \right\|_2^2 + \sum_{r \in \mathcal{R}_{i-1}} \omega(r) \left\| \mathbf{r}_i - \mathbf{r}_{i-1} \right\|_2^2, \tag{25}$$

where where $omega(x)$ is the regularization weight for $x$. The results are shown in Table 9. One can observe that the performance of $\text{CLKGE}_1$ is superior to $\text{CLKGE}_2$. In this way, it validates the rationality of our method and the effectiveness of CLKGE to model the association between true embedding and the obtained embedding for knowledge.

**Sensitivity Study.** We carry out a sensitivity study on the $\beta_1$ parameter in Eq.(9) in the text, in the original we only need to change the value of $\beta_1$ to study the sensitivity of the model since $\beta_1 + \beta_2 = 1$. We varied $\beta_1$ from 0-1 at intervals of 0.1. As shown in the Table 10 on ENTITY and

Table 6: Experiments for WN18RR-5-LS dataset.

| Models | MRR | H@1 | H@3 | H@10 |
|---|---|---|---|---|
| EMR | $.351_{\pm.002}$ | $.232_{\pm.002}$ | $.317_{\pm.003}$ | $.380_{\pm.001}$ |
| DiCGRL | $.365_{\pm.002}$ | $.244_{\pm.003}$ | $.325_{\pm.002}$ | $.392_{\pm.002}$ |
| LKGE | $.372_{\pm.002}$ | $.251_{\pm.002}$ | $.337_{\pm.003}$ | $.401_{\pm.001}$ |
| CLKGE | $.384_{\pm.002}$ | $.260_{\pm.002}$ | $.347_{\pm.003}$ | $.415_{\pm.002}$ |

Table 7: Statistical data of the four constructed growing KG datasets. For the i-th snapshot, $\mathcal{T}_{\Delta_i}$ denotes the set of new facts in this snapshot, and $\mathcal{E}_i$, $\mathcal{R}_i$ denote the sets of cumulative entities and relations in the first $i$ snapshots, respectively.

| Datasets | Snapshot 1 | | | Snapshot 2 | | | Snapshot 3 | | | Snapshot 4 | | | Snapshot 5 | | |
|---|---|---|---|---|---|---|---|---|---|---|---|---|---|---|---|
| | $|\mathcal{T}_{\Delta_1}|$ | $|\mathcal{E}_1|$ | $|\mathcal{R}_1|$ | $|\mathcal{T}_{\Delta_2}|$ | $|\mathcal{E}_2|$ | $|\mathcal{R}_2|$ | $|\mathcal{T}_{\Delta_3}|$ | $|\mathcal{E}_3|$ | $|\mathcal{R}_3|$ | $|\mathcal{T}_{\Delta_4}|$ | $|\mathcal{E}_4|$ | $|\mathcal{R}_4|$ | $|\mathcal{T}_{\Delta_5}|$ | $|\mathcal{E}_5|$ | $|\mathcal{R}_5|$ |
| ENTITY | 46,388 | 2,909 | 233 | 72,111 | 5,817 | 236 | 73,785 | 8,275 | 236 | 70,506 | 11,633 | 237 | 47,326 | 14,541 | 237 |
| RELATION | 98,819 | 11,560 | 48 | 93,535 | 13,343 | 96 | 66,136 | 13,754 | 143 | 30,032 | 14,387 | 190 | 21,594 | 14,541 | 237 |
| FACT | 62,024 | 10,513 | 237 | 62,023 | 12,779 | 237 | 62,023 | 13,586 | 237 | 62,023 | 13,894 | 237 | 62,023 | 14,541 | 237 |
| HYBRID | 57,561 | 8,628 | 86 | 20,873 | 10,040 | 102 | 88,017 | 12,779 | 151 | 103,339 | 14,393 | 209 | 40,326 | 14,541 | 237 |

Table 8: Cumulative time (seconds) cost on FACT during 5 snapshots.

| Models | Re-tarining | CWE | DiCGRL | PNN | GEM | EMR | EWC | SI | LKGE | Fine-tuning | CLKGE |
|---|---|---|---|---|---|---|---|---|---|---|---|
| | 5000 | 3100 | 2150 | 1900 | 1750 | 1540 | 1500 | 1400 | 1300 | 1350 | 1200 |

RELATION datasets with the MRR metric. When $\beta_1$ is 0, the model is degraded to use only the knowledge retention module, so the model performance decreases more; when $\beta_1$ is 1, the model is degraded to use only the knowledge migration module, so the model performance decreases more; this shows that the above two modules are very important. For the other parameters, it can be seen that the model performance does not change much, indicating that our model is not sensitive to the hyperparameters of the model.

Table 9: Ablation Study for Knowledge Retention.

| Model | ENTITY | | | | RELATION | | | |
|---|---|---|---|---|---|---|---|---|
| | MRR | H@1 | H@3 | H@10 | MRR | H@1 | H@3 | H@10 |
| CLKGE$_1$ | 0.248 | 0.144 | 0.278 | 0.436 | 0.203 | 0.115 | 0.226 | 0.379 |
| CLKGE$_2$ | 0.234 | 0.136 | 0.271 | 0.429 | 0.196 | 0.107 | 0.219 | 0.372 |

Table 10: Sensitivity Study for ENTITY and RELATION dataset.

| $\beta_1$ | 0 | 0.1 | 0.2 | 0.3 | 0.4 | 0.5 | 0.6 | 0.7 | 0.8 | 0.9 | 1.0 |
|---|---|---|---|---|---|---|---|---|---|---|---|
| ENTITY | .236 | .241 | .243 | .242 | .245 | .246 | .248 | .246 | .240 | .235 | .161 |
| RELATION | .172 | .194 | .199 | .201 | .203 | .204 | .203 | .198 | .194 | .192 | .137 |

