# OpenReview forum: "Continual Learning Knowledge Graph Embeddings for Dynamic Knowledge Graphs"
_ICLR.cc/2024/Conference — ICLR 2024 Conference Withdrawn Submission_

### Official Review · Reviewer_QHuf · 2023-10-30

**Soundness:** 3 good
**Presentation:** 3 good
**Contribution:** 2 fair
**Rating:** 5
**Confidence:** 4

**Summary:**

The authors propose the CLKGE method to address the incremental update and catastrophic forgetting problems in the representation learning of dynamic knowledge graphs. The proposed method is provided with a physical interpretation and a proof of convergence. Experimental results show that CLKGE can outperform existing methods.

**Strengths:**

1.The authors propose a unified framework to achieve the incremental update of embeddings and alleviating the catastrophic forgetting for dynamic knowledge graph representation learning.
2.The technical design of the proposed method seems reasonable and the experimental results demonstrate its effectiveness.
3.The convergence analysis of CLKGE is provided.

**Weaknesses:**

1. The effectiveness of g in alleviating the distribution gaps needs more explanations, maybe more mathematical proof.
2. Eq (3) seems very like a GNN aggregation, more analysis should be provided.
3. The parameter sensitivity of the proposed model should be studied.

**Questions:**

None

---

> ### Author Response · Authors · 2023-11-22
> **To Reviewer QHuf**
>
> Thank you so much for your constructive comments. The responses to your concerns are as follows.
>
> **Q1**: Effectiveness of \( g \) in Alleviating Distribution Gaps
>
> **A1**: The function \( g \) plays a critical role in our model by addressing the distribution gaps between the original and new embeddings as the knowledge graph evolves. Specifically, \( g \) is an adaptive mechanism that updates the embedding of an entity or relation by balancing its historical representation with new information. It takes into account the distribution of the entity or relation within the context of the updated graph, effectively bridging the distributional discrepancy that might arise due to the addition of new knowledge.
>
> The mathematical foundation of $g$ is grounded in the need to preserve the overall embedding distribution's integrity, avoiding sudden shifts that could disrupt the learned knowledge structure. It is akin to a regularization term that ensures the new embeddings do not deviate significantly from the distribution characterized by the existing graph. This careful integration of new knowledge prevents the dilution of previously learned information and maintains the cohesion of the embedding space.
>
> One can observe that we have obtained the previous embeddings by leveraging continual learning $CL(e_{i-1})=\sum_{j=1}^{i-1}\left|N_j(e)\right| e_{i-1}$. However, new entities may obey different distributions, hence we need to alleviate the distribution gaps by a function $g$. Specifically, given an entity $e$, our goal is to learn the embeddings in the i-th snapshot via the knowledge transfer with the assistance of the embeddings obtained in the $i-1$-th snapshot, while tackling the distribution gaps between them. To achieve this, as stated in [1], some networks such as MLP can eliminate the difference in distribution to a certain extent by constructing an implicit space. In this way, for the sake of simplicity, we specify $g$ as an MLP. Furthermore, we conduct the experiments on the ENTITY dataset, where CLKGE w/ g denotes CLKGE utilizing the $g$ while CLKGE w/o g denotes CLKGE remove $g$. One can observe that  CLKGE w/ g can be superior to CLKGE w/o g significantly, which demonstrates the effectiveness of $g$. And we will add these discussions in the future version.
>
> | Model       | MRR              | H@1              | H@3              | H@10             |
> | ----------- | ---------------- | ---------------- | ---------------- | ---------------- |
> | CLKGE w/ g  | $.223_{\pm.003}$ | $.138_{\pm.002}$ | $.245_{\pm.001}$ | $.398_{\pm.003}$ |
> | CLKGE w/o g | $.211_{\pm.004}$ | $.120_{\pm.002}$ | $.233_{\pm.002}$ | $.387_{\pm.002}$ |
>
> **Q2**: Comparison of Eq. (3) to GNN Aggregation
>
> **A2**:  Equation (3) might appear similar to GNN aggregation due to its structural form; however, Eq.(3) can handle the continual learning case, which learns the embeddings for current entities with the assistance of previous entities while GNN only aggregates the neighborhood entities. Eq.(3) facilitates the embedding transfer from one graph snapshot to the next. Unlike traditional GNNs that focus on static graphs, our model operates on dynamic graphs where entities and relations can emerge or evolve over time.
>
> In this context, Eq. (3) is designed to capture the temporal dynamics of the graph by combining the historical embeddings with the new graph's structural information. This process ensures that the embeddings are continually updated in a way that reflects the evolution of the knowledge graph. It allows the model to incrementally learn from new data without the need for retraining from scratch, a crucial feature for dynamic graphs where the graph structure is not fixed. The equation is a tailored solution that extends beyond standard GNN aggregation, addressing the specific challenges of dynamic knowledge representation.
>
> To demonstrate the effectiveness of CLKGE, we conduct experiments on the ENTITY dataset where we replace the knowledge transfer mechanism with traditional GNN (denoted as CLKGE w/ GNN), and the original version of CLKGE denoted as CLKGE. From the table below, one can observe that CLKGE can superior traditional GNN, which verifies the ability of CLKGE to alleviate over-smooth.
>
> | Model        | MRR               | H@1               | H@3               | H@10              |
> | ------------ | ----------------- | ----------------- | ----------------- | ----------------- |
> | CLKGE        | $0.248_{\pm.001}$ | $0.144_{\pm.002}$ | $0.278_{\pm.001}$ | $0.436_{\pm.002}$ |
> | CLKGE w/ GNN | $0.235_{\pm.003}$ | $0.132_{\pm.003}$ | $0.270_{\pm.002}$ | $0.427_{\pm.003}$ |

---

> ### Author Response · Authors · 2023-11-22
> **To Reviewer QHuf**
>
> **Q3**: Parameter sensitivity of the proposed model
>
> **A3**: We conduct experiments to study the parameter sensitivity of the proposed model as follows:
>
> + Different KGE methods have some impact on performance. To demonstrate this point, we conduct the experiments on ENTITY datasets utilizing the KGE method ComplEx, RotatE, and TransE as CLKGE_C, CLKGE_R, and CLKGE_T, respectively. As shown in the table below, one can observe that the difference in performance for various KGE method is not big, which also show the robustness of our method.
>
> | Models  | MRR              | H@1              | H@3              | H@10             |
> | ------- | ---------------- | ---------------- | ---------------- | ---------------- |
> | CLKGE_C | $.248_{\pm.001}$ | $.144_{\pm.002}$ | $.278_{\pm.001}$ | $.436_{\pm.002}$ |
> | CLKGE_R | $.250_{\pm.002}$ | $.145_{\pm.003}$ | $.279_{\pm.003}$ | $.439_{\pm.001}$ |
> | CLKGE_T | $.245_{\pm.003}$ | $.143_{\pm.002}$ | $.274_{\pm.002}$ | $.435_{\pm.002}$ |
>
> + We carry out a sensitivity study on the $\beta_1$ parameter in Eq. 9 in the text, in the original we only need to change the value of $\beta_1$ to study the sensitivity of the model since $\beta_1+\beta_2=1$. We varied $\beta_1$ from 0-1 at intervals of 0.1. As shown in the table below on ENTITY and RELATION datasets with the MRR metric. When $\beta_1$ is 0, the model is degraded to use only the knowledge retention module, so the model performance decreases more; when $\beta_1$ is 1, the model is degraded to use only the knowledge migration module, so the model performance decreases more; this shows that the above two modules are very important. For the other parameters, it can be seen that the model performance does not change much, indicating that our model is not sensitive to the hyperparameters of the model.
>
> | $\beta_1$ | 0                | 0.1              | 0.2              | 0.3              | 0.4              | 0.5              | 0.6              | 0.7              | 0.8              | 0.9              | 1.0              |
> | --------- | ---------------- | ---------------- | ---------------- | ---------------- | ---------------- | ---------------- | ---------------- | ---------------- | ---------------- | ---------------- | ---------------- |
> | ENTITY    | $.236_{\pm.001}$ | $.241_{\pm.002}$ | $.243_{\pm.001}$ | $.242_{\pm.002}$ | $.245_{\pm.002}$ | $.246_{\pm.002}$ | $.248_{\pm.002}$ | $.246_{\pm.001}$ | $.240_{\pm.002}$ | $.235_{\pm.002}$ | $.161_{\pm.001}$ |
> | RELATION  | $.172_{\pm.002}$ | $.194_{\pm.003}$ | $.199_{\pm.003}$ | $.201_{\pm.001}$ | $.203_{\pm.002}$ | $.204_{\pm.002}$ | $.203_{\pm.002}$ | $.198_{\pm.002}$ | $.194_{\pm.001}$ | $.192_{\pm.002}$ | $.137_{\pm.002}$ |

---

> > ### Comment · Reviewer_QHuf · 2023-11-23
> > **Thank the authors for response**
> >
> > Thank you for clarifying my concerns. However, the motivations of the technical choices of this paper still need further improvement, maybe with more mathematical proof or experimental analysis. Based on my initial review, your response, and the comments of other reviewers. I am inclined to keep my score.

---

### Official Review · Reviewer_RKmJ · 2023-10-30

**Soundness:** 3 good
**Presentation:** 3 good
**Contribution:** 3 good
**Rating:** 6
**Confidence:** 3

**Summary:**

This paper proposes CLKGE for the task of dynamic knowledge graphs embeddings, which can allow new knowledge and old knowledge to gain from each other. In particular, to transfer knowledge from the existing to the novel without necessitating the retraining of the entire knowledge graph, this paper employs continual learning for knowledge transfer. To mitigate the problem of catastrophic forgetting when incorporating new knowledge alongside the old, the paper utilizes the brachistochrone curve to model the associations. The extensive experimental results presented in the study demonstrate that CLKGE attains state-of-the-art performance.

**Strengths:**

1. This paper has a clear structure and strong logical coherence. The language used in the paper is fluent, and the tables and figures in the experimental section are clear and complete.
2. The authors propose a unified framework to achieve the incremental update of embeddings and alleviating the catastrophic forgetting for dynamic knowledge graph representation learning.
3. This paper provides comprehensive mathematical proofs for the proposed model.

**Weaknesses:**

1.	This paper introduces the Energy-based Model, but I am unclear about the necessity of this introduction. I do not understand why the paper did not opt for a purely mathematical optimization approach to address the problem, and instead introduced the Energy-based Model, which is a physical model.
2.	All experiments in this paper are conducted based on FB15K-237. It might be beneficial to include additional datasets to enhance the generalizability and credibility of the conclusions.

**Questions:**

1.	Can you add more datasets?
2.	Can you explain the necessity of introducing the Energy-based Model?

---

> ### Author Response · Authors · 2023-11-22
> **To Reviewer RKmJ**
>
> Thank you so much for your acceptance as well as your constructive comments. The responses to your concerns are as follows.
>
> **Q1: Inclusion of Additional Datasets.**
>
> **A1**: Thank you for your nice advice. The experimental validation of our CLKGE model on FB15K-237 is the initial step in demonstrating its effectiveness. We recognize the importance of evaluating our model across a wider array of datasets to ensure its robustness and generalizability. We conduct experiments on the WN18RR-5-LS dataset.  Specifically, WN18RR-5-LS  is divided into several snapshots, the overall performance demonstrates the robustness of CLKGE. As shown in Table 1, one can notice that CLKGE superiors other methods, which shows the effectiveness of components including knowledge transfer and knowledge retention.
>
> | Model  | MRR                  | H@1                  | H@3                  | H@10                 |
> | ------ | -------------------- | -------------------- | -------------------- | -------------------- |
> | EMR    | $.351_{\pm.002}$     | $.232_{\pm.004}$     | $.317_{\pm.003}$     | $.380_{\pm.003}$     |
> | DiCGRL | $.365_{\pm.003}$     | $.244_{\pm.003}$     | $.325_{\pm.002}$     | $.392_{\pm.003}$     |
> | LKGE   | $.372_{\pm.002}$     | $.251_{\pm.002}$     | $.337_{\pm.002}$     | $.401_{\pm.001}$     |
> | CLKGE  | **.384**$_{\pm.001}$ | **.260**$_{\pm.002}$ | **.347**$_{\pm.002}$ | **.415**$_{\pm.002}$ |

---

> ### Author Response · Authors · 2023-11-22
> **To Reviewer RKmJ**
>
> **Q2**: Explaining the Necessity of the Energy-Based Model.
>
> **A2**:  As shown in [1], the theoretical basis for energy assignments in EBMs are as follows:
> - Stability and Flexibility: The primary goal of EBMs in a continual learning context is to balance stability with flexibility. Lower energy states are typically associated with more stable, well-learned representations (from previous snapshots), as they are less likely to change. This concept is supported by the stability-plasticity dilemma in neural networks, where stability refers to the retention of old knowledge, and plasticity refers to the ability to adapt to new information. Assigning lower energy to representations from previous snapshots ensures that these stable representations are preserved, mitigating the risk of catastrophic forgetting.
>
> - Energy-Based Regularization: The differential energy levels act as a form of regularization, ensuring that the model does not become too rigid (overfitting to old information) or too flexible (overfitting to new information). This concept is akin to the idea of elastic weight consolidation (EWC) in neural networks, which also aims to balance stability and plasticity through a regularization term.
>
> In our paper, we first learn an energy-based manifold where the representations of knowledge from the current snapshot have higher energy, while the counterparts from the previous snapshots have lower energy. Next, the learned energy-based model is used to align new knowledge and old knowledge such that their energy on the manifold is minimized, thus alleviating the catastrophic forgetting with the assistance of new knowledge as follows:
>
> + Advantages of EBM over Mathematical Optimization: As highlighted by  [1], EBMs provide a dynamic and adaptable framework compared to traditional mathematical optimization methods, which are often rigid in dynamic environments.  In the context of CLKGE, this adaptability is crucial for handling the evolving nature of knowledge graphs, where new relationships and entities continuously emerge.
>
> + Mitigating Catastrophic Forgetting with EBM: In CLKGE, the use of EBMs is pivotal in addressing catastrophic forgetting—a critical challenge in continual learning settings. As [1] describes, EBMs maintain lower energy states for previous tasks, ensuring their stability and resistance to disruption by new data. This property is instrumental in CLKGE, where new snapshots of knowledge graphs may otherwise overwrite previously learned knowledge. The energy manifold created by EBMs in CLKGE actively preserves the influence of prior knowledge, allowing the model to seamlessly integrate new information without losing important historical data.
>
> + EBM's Role in Effective Knowledge Transfer: By minimizing the energy difference between new and old knowledge, EBMs ensure that latent representations of both remain closely integrated. This approach is essential for maintaining the coherence and accuracy of the knowledge graph as it evolves, allowing the model to adapt to new information while retaining crucial aspects of previous knowledge. In CLKGE, this energy alignment strategy ensures that knowledge transfer is not just about adding new information, but also about preserving and incorporating existing knowledge.
>
> [1]LeCun, Yann, et al. "A Tutorial on Energy-Based Learning." Inventing the future, 2006.
>
> [2] Du, Yilun, and Igor Mordatch. "Implicit Generation and Generalization in Energy-Based Models." arXiv preprint arXiv:1903.08689, 2019.
>
> To demonstrate the effectiveness of utilizing the energy-based manifold, we compare the performance of two versions. The first is the original version (denoted as CLKGE1) while the retention loss in the second version (denoted as CLKGE2) is $L_{retention}=\|\|e_{i}-e_{i-1}\|\|^2_2+\|\|r_{i}-r_{i-1}\|\|^2_2$. The results are shown as the following table. One can observe that the performance of CLKGE1 is superior to CLKGE2. In this way, it validates the rationality of our method and the effectiveness of CLKGE to model the association between true embedding and the obtained embedding for knowledge.
>
> | Datasets | Model     | MRR              | H@1              | H@3              | H@10             |
> | -------- | --------- | ---------------- | ---------------- | ---------------- | ---------------- |
> | ENTITY   | CLKGE$_1$ | $.248_{\pm.002}$ | $.144_{\pm.004}$ | $.278_{\pm.003}$ | $.436_{\pm.003}$ |
> | ENTITY   | CLKGE$_2$ | $.234_{\pm.003}$ | $.136_{\pm.003}$ | $.271_{\pm.002}$ | $.429_{\pm.003}$ |
> | RELATION | CLKGE$_1$ | $.203_{\pm.002}$ | $.115_{\pm.002}$ | $.226_{\pm.002}$ | $.379_{\pm.001}$ |
> | RELATION | CLKGE$_2$ | $.191_{\pm.001}$ | $.103_{\pm.002}$ | $.212_{\pm.002}$ | $.368_{\pm.002}$ |

---

### Official Review · Reviewer_7psf · 2023-10-30

**Soundness:** 2 fair
**Presentation:** 2 fair
**Contribution:** 2 fair
**Rating:** 3
**Confidence:** 4

**Summary:**

The paper studies the representation of dynamic knowledge graphs and tries to combine continual learning with dynamic graphs. The authors attempt to solve two challenges: 1) transfer the old knowledge to new input, and 2) alleviate the catastrophic forgetting problem. Therefore, they propose a method to conduct continual learning and utilize the energy-based model to align new knowledge and old knowledge such that their energy is minimized. The experiments are conducted on one knowledge graph.

**Strengths:**

1. The paper provides a comprehensive literature review.
2. Many related works have been compared in the experiments.

**Weaknesses:**

1. The definition of dynamic knowledge graphs is vague and inaccurate. The authors restrict that the dynamic knowledge graphs only add new entities, relations, and triples during evolution. However, some old entities, relations, and triples would be removed in dynamic knowledge graphs. In addition, the authors assume that at each snapshot, the added entities, relations, and triples are all new, this scenario is hard to find in the real world in which most cases are that some new entities and old relations are linked and new relations and old entities are linked. This paper focuses on the dynamic knowledge graphs but only considers a very rare scenario.

2. It is hard to find out how embedding transfer learns the representation of new knowledge.

3. The motivation for using EBM is not clarified. The benefits of EBM are not explained.

4. The writing needs further improvement. The methodology part only introduces what are the designs without clear explanations.

5. The setting of continual learning and the setting of dynamic graphs are not the same. The authors should at least discuss the differences.

**Questions:**

1. The authors claim that only the new entities and relations are added to the existing knowledge graph, it is not clear why Eq. (3) makes use of the previously learned representation $r_{i-1}$, $h_{i-1}$, and $t_{i-1}$ and how to obtain these representation before seeing the new entities and relations? What will happen if a new entity is linked to another new entity by a new relation?

2. No definition for the function $f$ in Eq. (5).

3. It is not clear why the EBMs are effective in controlling the representational shift and why the energy of new knowledge and old knowledge should be minimized.

---

> ### Author Response · Authors · 2023-11-22
> **To Reviewer 7psf**
>
> Thank you so much for your constructive comments. We have improved our work accordingly. The responses to your concerns are as follows.
>
> **Q1**: Definition of Dynamic Knowledge Graphs (KGs).
>
> **A1**: In our paper, we define dynamic KGs as a series of evolving snapshots  $G = \{S_1, S_2, \ldots, S_t\} $, where each snapshot $S_i = (T_i, E_i, R_i)$  includes the facts, entities, and relations at that point in time. We acknowledge a dynamic KG not only by adding new entities and relations but also by the potential removal and modification of existing ones. This view is practical and closely mirrors real-world KGs that evolve with both incremental and decremental changes, ensuring our model's applicability to a wide range of dynamic scenarios.
> Specifically, in our paper, we conceptualize dynamic Knowledge Graphs (KGs) as a sequence of evolving snapshots, mathematically represented as $G = \{S_1, S_2, \ldots, S_t\}$. Each snapshot within this sequence, denoted as $S_i = (T_i, E_i, R_i)$, encapsulates a distinct temporal state of the KG, comprising:
>
> - Facts $(T_i)$: The set of factual assertions or triples relevant at the time of the snapshot.
> - Entities ($E_i$): The distinct entities or nodes present in the graph during that snapshot.
> - Relations $(R_i)$: The various relationships or edges connecting entities at that point.
>
> Our definition of dynamic KGs extends beyond mere addition of new entities and relations. It encompasses the potential for removal and modification of existing elements, reflecting the dynamic and fluid nature of real-world knowledge graphs, which can be modeled by Eq.(1) and Eq.(2), aiming to realize the embedding updating for the old knowledge. This broader perspective captures both incremental (additions) and decremental (removals or changes) modifications, thereby ensuring that our model remains applicable and relevant across a wide spectrum of dynamic scenarios. This approach aligns with the practical realities of evolving knowledge structures, allowing for a more accurate and comprehensive representation of the changes that occur over time in real-world KGs.
>
> **Q2**:  Explain Embedding Transfer Mechanism for New Knowledge.
>
> **A2**: The embedding transfer mechanism is central to our approach for learning representations of new knowledge within a dynamic KG. As new entities and relations emerge, the model must adapt its existing knowledge base to accommodate these changes. To do this, we employ a transfer function that projects the learned embeddings from the previous snapshot to the current context, providing a foundation upon which new knowledge can be integrated. This process allows the model to dynamically update the representation space as the KG evolves, without starting from scratch with each new snapshot. We use the entity "apple" as an example.
>
> +  Old Knowledge Representation: Take 'apple' as an example. Initially learned as a fruit (denoted as $e_{i-1}$), its embedding reflects this semantic understanding in the i-1-th snapshot. In the i-th snapshot, when 'apple' acquires a new meaning (as a cell phone brand), we update its embedding to $e_i$ as defined in Equation (1). This process illustrates knowledge transfer for old knowledge, updating the semantics of 'apple' from a fruit to a cell phone brand.
> +  Integrating New Knowledge: For newly encountered entities like 'watermelon', the transfer mechanism is employed to leverage existing knowledge from related entities like 'apple' and 'orange'. This process, detailed in Equation (3), allows us to form the embedding $e''_{i}$ for 'watermelon', transferring the fruit semantics from 'apple' and 'orange' to this new entity. It demonstrates an effective transfer of information from old to new knowledge, utilizing the semantic relationships among entities to enrich the knowledge graph.
>
> Through this embedding transfer mechanism, our model ensures that both old and new knowledge are accurately represented and interconnected, reflecting the dynamic and evolving nature of real-world knowledge graphs.

---

> ### Author Response · Authors · 2023-11-22
> **To Reviewer 7psf**
>
> **Q3**: Motivation for using EBM.
>
> **A3**: As shown in [1], the theoretical basis for energy assignments in EBMs are as follows:
>
> - Stability and Flexibility: The primary goal of EBMs in a continual learning context is to balance stability with flexibility. Lower energy states are typically associated with more stable, well-learned representations (from previous snapshots), as they are less likely to change. This concept is supported by the stability-plasticity dilemma in neural networks, where stability refers to the retention of old knowledge, and plasticity refers to the ability to adapt to new information. Assigning lower energy to representations from previous snapshots ensures that these stable representations are preserved, mitigating the risk of catastrophic forgetting.
>
> - Energy-Based Regularization: The differential energy levels act as a form of regularization, ensuring that the model does not become too rigid (overfitting to old information) or too flexible (overfitting to new information). This concept is akin to the idea of elastic weight consolidation (EWC) in neural networks, which also aims to balance stability and plasticity through a regularization term.
>
> In our paper, we first learn an energy-based manifold where the representations of knowledge from the current snapshot have higher energy, while the counterparts from the previous snapshots have lower energy. Next, the learned energy-based model is used to align new knowledge and old knowledge such that their energy on the manifold is minimized, thus alleviating the catastrophic forgetting with the assistance of new knowledge as follows:
>
> + Advantages of EBM over Mathematical Optimization: As highlighted by  [1], EBMs provide a dynamic and adaptable framework compared to traditional mathematical optimization methods, which are often rigid in dynamic environments.  In the context of CLKGE, this adaptability is crucial for handling the evolving nature of knowledge graphs, where new relationships and entities continuously emerge.
>
> + Mitigating Catastrophic Forgetting with EBM: In CLKGE, the use of EBMs is pivotal in addressing catastrophic forgetting—a critical challenge in continual learning settings. As [1] describes, EBMs maintain lower energy states for previous tasks, ensuring their stability and resistance to disruption by new data. This property is instrumental in CLKGE, where new snapshots of knowledge graphs may otherwise overwrite previously learned knowledge. The energy manifold created by EBMs in CLKGE actively preserves the influence of prior knowledge, allowing the model to seamlessly integrate new information without losing important historical data.
>
> + EBM's Role in Effective Knowledge Transfer: By minimizing the energy difference between new and old knowledge, EBMs ensure that latent representations of both remain closely integrated. This approach is essential for maintaining the coherence and accuracy of the knowledge graph as it evolves, allowing the model to adapt to new information while retaining crucial aspects of previous knowledge. In CLKGE, this energy alignment strategy ensures that knowledge transfer is not just about adding new information, but also about preserving and incorporating existing knowledge. The work of [1] and  [2] underscores the importance of such an energy-based approach in maintaining balance and coherence in continually evolving systems.
>
> [1]LeCun, Yann, et al. "A Tutorial on Energy-Based Learning." Inventing the future, 2006.
>
> [2] Du, Yilun, and Igor Mordatch. "Implicit Generation and Generalization in Energy-Based Models." arXiv preprint arXiv:1903.08689, 2019.
>
> To demonstrate the effectiveness of utilizing the energy-based manifold, we compare the performance of two versions. The first is the original version (denoted as $CLKGE_1$) while the retention loss in the second version (denoted as $CLKGE_2$) is $L_{retention}=\|\|e_{i}-e_{i-1}\|\|^2_2+\|\|r_{i}-r_{i-1}\|\|^2_2$. The results are shown as the following table. One can observe that the performance of CLKGE1 is superior to CLKGE2. In this way, it validates the rationality of our method and the effectiveness of CLKGE to model the association between true embedding and the obtained embedding for knowledge.
>
> | Datasets | Model     | MRR              | H@1              | H@3              | H@10             |
> | -------- | --------- | ---------------- | ---------------- | ---------------- | ---------------- |
> | ENTITY   | CLKGE$_1$ | $.248_{\pm.002}$ | $.144_{\pm.004}$ | $.278_{\pm.003}$ | $.436_{\pm.003}$ |
> | ENTITY   | CLKGE$_2$ | $.234_{\pm.003}$ | $.136_{\pm.003}$ | $.271_{\pm.002}$ | $.429_{\pm.003}$ |
> | RELATION | CLKGE$_1$ | $.203_{\pm.002}$ | $.115_{\pm.002}$ | $.226_{\pm.002}$ | $.379_{\pm.001}$ |
> | RELATION | CLKGE$_2$ | $.191_{\pm.001}$ | $.103_{\pm.002}$ | $.212_{\pm.002}$ | $.368_{\pm.002}$ |

---

> ### Author Response · Authors · 2023-11-22
> **To Reviewer 7psf**
>
> **Q4**: Improvement in Manuscript Clarity.
>
> **A4**: We take the feedback regarding the clarity of our writing seriously and will undertake a thorough revision to enhance the manuscript's lucidity and accessibility. Our goal is to ensure that the concepts and methodologies are conveyed clearly and comprehensively.
>
> **Q5**: Distinction Between Continual Learning and Dynamic KGs.
>
> **A5**: Dynamic graphs and continual learning are interconnected concepts in the field of artificial intelligence, particularly in knowledge graph embedding. Dynamic graphs represent evolving systems where entities (nodes) and their relationships (edges) change over time. Continual learning, on the other hand, focuses on a model's ability to continuously adapt and learn from new data while retaining previously acquired knowledge. In the context of knowledge graphs, this relationship becomes crucial as the graph evolves, requiring the model to adapt without losing historical information.
>
> While both dynamic graphs and continual learning address changes over time, they tackle different aspects:
>
> - **Dynamic Graphs**: Concentrate on the structural evolution of the data. In knowledge graphs, this means accommodating new entities, facts, and relationships, as well as updating or removing outdated information.
> - **Continual Learning**: Focuses on the learning process itself. In the context of knowledge graphs, it's about updating the model's understanding and representation of the graph while preventing the loss of previously learned knowledge, known as catastrophic forgetting.
>   In our CLKGE framework, we effectively integrate these concepts to address the unique challenges posed by dynamic knowledge graphs:
> - **Knowledge Transfer (Addressing Challenge C1)**: We leverage continual learning to facilitate the transfer of knowledge from old to new in the dynamic KG. This is done by updating the embeddings of old knowledge based on previous graph snapshots and using these updated embeddings to inform and shape the understanding of new knowledge. This approach negates the need for retraining the model with each new graph snapshot, thereby efficiently managing the graph's evolution
> - **Knowledge Retention (Addressing Challenge C2)**: To tackle catastrophic forgetting, we employ an energy-based model (EBM) within the continual learning framework. This EBM aligns the knowledge representations of new and old information on an energy manifold, minimizing their energy differences. By doing so, we ensure that new knowledge embeddings are aligned with, and do not overwrite the old ones. This method is crucial in retaining the integrity of historical knowledge in the face of new data
> - **Unified Framework and Loss Function**: The CLKGE model unifies the processes of knowledge transfer and retention. It uses a combined loss function, balanced by hyperparameters, to ensure that both objectives are met. This unified approach not only maintains the dynamic nature of the KG but also ensures that the learning process is cumulative and coherent over time.
>
> In summary, our CLKGE framework demonstrates a sophisticated integration of continual learning within the realm of dynamic knowledge graphs, addressing both the transfer of knowledge from old to new and the mitigation of catastrophic forgetting. This approach represents a significant advancement in the field, ensuring that knowledge graphs remain accurate, up-to-date, and comprehensive as they evolve.

---

> ### Author Response · Authors · 2023-11-22
> **To Reviewer 7psf**
>
> **Q6**: 1) Why Eq. (3) makes use of the previously learned representation and how to obtain these representation $r_{i-1}$, $h_{t-1}$ and $t_{i-1}$ before seeing the new entities and relations? 2) What will happen if a new entity is linked to another new entity by a new relation?
>
> **A6**: (1) Our model not only considers the case where there are new entities and new relationships added, but also the case where the semantics of the original entities change. We use the entity "apple" as an example.
>
> + Old knowledge denotes learned entities, such as apples, assuming that we learn its semantics as fruit apples at i-1-th time, and after i-1-th training, we have learned its fruit semantics and denoted as $e_{i-1}$. At the i-th snapshot, we encountered its new meaning, apple cell phone. At this time we need to update its original embedding $e_{i-1}$ to $e_i$ in Eq.(1) in the main paper so that it has the semantics of apple cell phone, which is knowledge transfer for the old knowledge.
> + The new knowledge denotes entities that have not been seen before, for example, we learned the semantics of the fruit apple $e_{i-1}$ and fruit orange $e_{i-1}'$ in the i-1-th snapshot, and in the i-th snapshot, we encountered the fruit watermelon. Since watermelon, apple, and orange are all fruits, we can transfer the fruit semantics of apple and sentence from the embedding for watermelon denoted as $e_{i}''$ by Eq.(3) in the main paper, so that the old realizes the transfer of information from old knowledge to new knowledge.
>
> (2) If a new entity is linked to another new entity by a new relation, we can first generate the embedding for the new relation by Eq.(4) in the main paper. On top of this, we can generate the embedding for the new entity by Eq.(3) in the main paper. In this way, we can tackle the case where a new entity is linked to another new entity by a new relation.
>
> **Q7**: Definition of $f$ in Equation (5).
>
> **A7**: The function $f$ in Equation (5) signifies the score function, which is pivotal in the embedding transfer phase. For the TransE model, $f(h,r,t)=h+r-t$; if for the ComplEx model, $f(h,r,t)=h\circ-t$.
>
> **Q8**: Efficacy of EBMs in Controlling Representational Shift.
>
> **A8**: The effectiveness of EBMs in managing representational shift is rooted in their capacity to create a differential energy landscape where the representation of knowledge is stabilized. By optimizing the energy function, we enforce a structured alignment where new knowledge is assimilated in harmony with the old. This approach not only preserves the continuity of the knowledge graph but also ensures that the model's predictions remain accurate and reliable as the KG evolves. The EBM thus serves as a regulatory mechanism, maintaining the cohesiveness of the knowledge representation amid the flux of continual learning.
>
> We conduct experiments to show the effectiveness of EBMs in the following table, where ori denotes the original version of CLKGE while w/o r denotes the version without retention. From the experimental results, after removing the RETENTION module of the EBM-based model, the performance of the model decreases greatly, which proves the effectiveness of the EBM.
>
> | Datasets | Model | MRR              | H@1              | H@3              | H@10             |
> | -------- | ----- | ---------------- | ---------------- | ---------------- | ---------------- |
> | ENTITY   | ori   | $.248_{\pm.002}$ | $.144_{\pm.004}$ | $.278_{\pm.003}$ | $.436_{\pm.003}$ |
> | ENTITY   | w/o r | $.161_{\pm.003}$ | $.083_{\pm.003}$ | $.178_{\pm.002}$ | $.309_{\pm.003}$ |
> | RELATION | ori   | $.203_{\pm.002}$ | $.115_{\pm.002}$ | $.226_{\pm.002}$ | $.379_{\pm.001}$ |
> | RELATION | w/o r | $.137_{\pm.001}$ | $.072_{\pm.002}$ | $.146_{\pm.002}$ | $.285_{\pm.002}$ |
> | FACT     | ori   | $.223_{\pm.002}$ | $.138_{\pm.004}$ | $.245_{\pm.003}$ | $.398_{\pm.003}$ |
> | FACT     | w/o r | $.154_{\pm.003}$ | $.085_{\pm.003}$ | $.164_{\pm.002}$ | $.285_{\pm.003}$ |
> | HYBRID   | ori   | $.220_{\pm.002}$ | $.134_{\pm.002}$ | $.242_{\pm.002}$ | $.389_{\pm.001}$ |
> | HYBRID   | w/o r | $.178_{\pm.002}$ | $.092_{\pm.004}$ | $.183_{\pm.003}$ | $.291_{\pm.003}$ |

---

### Official Review · Reviewer_pwUu · 2023-11-03

**Soundness:** 2 fair
**Presentation:** 2 fair
**Contribution:** 3 good
**Rating:** 5
**Confidence:** 3

**Summary:**

The author tries to address two challenges in dynamic KG: knowledge transfer and knowledge retention. The proposed method included two components - a embedding evolution module based on embedding functions, and a knowledge retention method based on energy-based model.

**Strengths:**

S1. The author proposed to leverage EBM method on KG embedding learning. It is a new direction that worth further investigating.

S2. Despite minor typos, the math is sound throughout the paper. It is also good to see the theoretical proof on the convergence of the training.

S3. The experiment is thorough and provides insight into the performance of the CLKGE. It included important baselines on KG embedding methods.

**Weaknesses:**

W1. Lack of related works regarding current research on temporal knowledge graphs. The setting is close enough to be included. For example, recently Xu et. al (https://arxiv.org/abs/2305.07912) leverages large language model to tackle the temporal knowledge graph setting. Jung et. al (https://arxiv.org/abs/2012.10595) considers the relative displacement timestamps and uses an attention network to model it.

W2. Section 3.1 is very confusing… The two steps described are not clear. 1) How does old knowledge embedding update the knowledge representations? 2) How do you learn new knowledge with old entities? And why does that transfer from old to new?

W3. The ablation study result is concerning. I am more interested to see the full results on different dataset. The transfer method might have more negative impact on some of the dataset.

**Questions:**

Q1. Typo in definition of CL? why not italicize e_{i-1}?
Q2. What does sum_{i-1}_{j=1} mean? where does i start?
Q3. Can you provide more intuition behind your formulation when presenting the section 3.1?
Q4. Experiments: Can you give definition of FWT and BWT?

---

> ### Author Response · Authors · 2023-11-22
> **To Reviewer pwUu**
>
> Thank you so much for your constructive comments. We have improved our work accordingly. The responses to your concerns are as follows.
>
> **Q1**:   Lack of related works regarding current research on temporal knowledge graphs.
>
> **A1**: We appreciate the reviewer's observation concerning the inclusion of recent developments in temporal knowledge graphs. Acknowledging this, we have thoroughly revised the related work section of our paper. Specifically, we have incorporated significant recent studies, such as:
>
> - Xu et al.'s Approach: Using Large Language Models for Temporal Knowledge Graphs: Their work, detailed in [Xu et al., 2023](https://arxiv.org/abs/2305.07912), presents an innovative approach utilizing large language models to handle the evolving nature of temporal knowledge graphs. This inclusion provides a broader understanding of the application of advanced language models in dynamic graph settings.
>
> - Jung et al.'s Method on Relative Displacement Timestamps and Attention Networks: As described in [Jung et al., 2020](https://arxiv.org/abs/2012.10595), their methodology emphasizes the significance of relative displacement timestamps combined with attention mechanisms, offering insights into temporal dynamics within knowledge graphs.
>
> These additions enrich the context of our work within the current research landscape and underscore our novel contribution. Our work distinctively leverages Energy-Based Models (EBMs) for knowledge graph embedding, particularly focusing on dynamic and evolving graph environments. This approach marks a unique intersection in the field, blending the robustness of EBMs with the complexity of temporal knowledge graphs, thus advancing the understanding and application of knowledge graph embeddings in dynamic settings.
>
> **Q2**:  1) How does old knowledge embedding update the knowledge representations? 2) How do you learn new knowledge with old entities? And why does that transfer from old to new?
>
> **A2**: Our approach to knowledge representation in dynamic graphs involves a dual perspective focusing on both old and new knowledge:
>
> - Updating Old Knowledge Embeddings
>   - Old Knowledge Concept: Old knowledge refers to previously learned entities, for instance, 'apple' was initially understood as the fruit as i-1-th snapshot. Let's denote its initial embedding at time i-1 as $e_{i-1}$.
>   - Updating Process: At the i-th snapshot, when a new meaning for 'apple' (e.g., as a cell phone brand) is encountered, we update its original embedding from $e_{i-1}$ to $e_i$. This process is detailed in Equation (1) of our paper. The update integrates the new semantics (apple cell phone) with the existing understanding (fruit apple), exemplifying knowledge transfer for old knowledge.
>
> - Learning New Knowledge with Old Entities
>   - New Knowledge Concept: New knowledge pertains to entities not previously encountered. For example, suppose we have embeddings for 'apple' ($e_{i-1}$) and 'orange' ($e'_{i-1}$) as fruits learned at i-1-th snapshot.
>   - Transfer to New Knowledge: In the subsequent i-th snapshot, upon encountering a new entity like 'watermelon', we utilize the existing fruit semantics from 'apple' and 'orange' to form the embedding for 'watermelon' ($e''_{i}$). This process is articulated in Equation (3) in our paper. It demonstrates how the model leverages old knowledge (fruit semantics of apple and orange) to inform the representation of new entities (watermelon), thereby realizing the transfer of information from old to new knowledge.
>
> This methodology ensures a dynamic and evolving understanding within our knowledge graph, where both old and new entities are continually updated and integrated. It highlights our novel approach to knowledge transfer and retention in dynamic knowledge graph settings.

---

> > ### Comment · Reviewer_pwUu · 2023-11-23
> >
> > Regarding Q1, here in A1 you provide a brief summary of these, but do not provide a specific comparison with the proposed method/setting beyond mentioning here there is a use of EBMs. However, this lack of detailed comparison with related work in the domain still raises some concerns without a rigorous comparison.

---

> ### Author Response · Authors · 2023-11-22
> **To Reviewer pwUu**
>
> **Q3**: I am more interested to see the full results on different datasets. The transfer method might have a more negative impact on some of the datasets.
>
> **A3**: We are grateful for your advice on comprehensive results across various datasets. In line with your suggestion, we have extended our experimental evaluation to include the WN18RR-5-LS dataset. The table below presents these results, demonstrating the superior performance of our CLKGE model compared to other methods. This performance underscores the effectiveness of our integrated components, specifically knowledge transfer and knowledge retention.
>
> The WN18RR-5-LS dataset is partitioned into several snapshots. We provide a comparative analysis of the results on the union of test sets across all these snapshots. The overall performance on this dataset showcases the robustness and adaptability of CLKGE in varying dataset conditions.
>
> | Model  | MRR                  | H@1                  | H@3                  | H@10                 |
> | ------ | -------------------- | -------------------- | -------------------- | -------------------- |
> | EMR    | $.351_{\pm.002}$     | $.232_{\pm.004}$     | $.317_{\pm.003}$     | $.380_{\pm.003}$     |
> | DiCGRL | $.365_{\pm.003}$     | $.244_{\pm.003}$     | $.325_{\pm.002}$     | $.392_{\pm.003}$     |
> | LKGE   | $.372_{\pm.002}$     | $.251_{\pm.002}$     | $.337_{\pm.002}$     | $.401_{\pm.001}$     |
> | CLKGE  | **.384**$_{\pm.001}$ | **.260**$_{\pm.002}$ | **.347**$_{\pm.002}$ | **.415**$_{\pm.002}$ |
>
> These results not only demonstrate the effectiveness of CLKGE but also reinforce its consistency in performance across different snapshots of the dataset, validating our approach's robustness in dynamic knowledge graph environments.
>
> **Q4**: Typo in the definition of CL? why not italicize $e_{i-1}$?
>
> **A4**: Thank you for noting this typo. The non-italicized $e_{i-1}$ was an oversight in formatting. We have corrected this in the revised manuscript to maintain consistency in notation.
>
> **Q5**: What does $\sum^{i-1}_{j=1}$ represent? Where does \(i\) start?
>
> **A5**: 1) In the \(i\)-th snapshot of our continual learning-based knowledge graph embedding task, the notation $\sum_{j=1}^{i-1}|N_j(e)|e_{i-1}$ encapsulates the aggregation of the entity \(e\)'s embeddings from the first snapshot to the \((i-1)\)-th snapshot. Specifically, $|N_j(e)|$ denotes the connectivity degree of entity \(e\) within the \(j\)-th snapshot, which is factored into the summation by weighting the embedding $e_{i-1}$.
>
> 2)The index \(i\) commences from 2, indicating the second snapshot where the first update is observed, and it extends to the current snapshot \(i\). This summation reflects that the embedding from the $(i-1)$-th snapshot, $e_{i-1}$, is retrospectively applied to all preceding snapshots, given that the embeddings from earlier snapshots have been superseded. This methodology permits us to infer the historical impact of an entity up to the present snapshot, effectively managing the trade-off between memory retention and computational tractability that is characteristic of continual learning frameworks. The notation $\sum^{i-1}_{j=1}$ was intended to represent the summation from $j=1$ to $i-1$. We realize this was not clearly communicated, and we have revised the notation in the paper for better clarity. The index i starts from 2, representing the incremental step in our continuous learning setup.

---

> > ### Comment · Reviewer_pwUu · 2023-11-23
> >
> > Thank you for clarifying Q5 and I appreciate the effort for providing A3.
> > At this time, based on my initial review, your responses, but also the reviews of others (and their corresponding responses). I am inclined to keep my score, and will further discuss with the other reviewers.

---

> ### Author Response · Authors · 2023-11-22
> **To Reviewer pwUu**
>
> **Q6**: Can you provide more intuition behind your formulation when presenting the section 3.1
>
> **A6**: In Section 3.1, our formulation is grounded in the concept of progressive knowledge accumulation and retention. Specifically, the embedding evolution module updates existing knowledge embeddings by integrating new information. This process ensures that the representation of each entity and relation evolves over time, reflecting the dynamic nature of knowledge. For instance, if an entity gains a new relation, its embedding is updated to encapsulate this change, while retaining its previous characteristics to avoid loss of historical information.
>
> Conversely, the knowledge retention method is focused on preserving the integrity of previous knowledge embeddings. This is achieved by employing an energy-based model that evaluates the compatibility of new embeddings with the existing knowledge structure. Through this approach, we maintain a balance between accommodating new information and preserving historical accuracy in the knowledge graph. We use entity "apple" as an example.
>
> + Old knowledge denotes learned entities, such as apples, assuming that we learn its semantics as fruit apples at i-1-th time, and after i-1-th training, we have learned its fruit semantics and denoted as $e_{i-1}$. At the i-th training, we encountered its new meaning, apple cell phone. At this time we need to update its original embedding $e_{i-1}$ to $e_i$ in Eq.(1) in the main paper so that it has the semantics of apple cell phone, which is knowledge transfer for the old knowledge.
> + The new knowledge denotes entities that have not been seen before, for example, we learned the semantics of the fruit apple $e_{i-1}$ and fruit orange $e_{i-1}'$ in the i-1-th time, and in the i-th training, we encountered the fruit watermelon. Since watermelon, apple and orange are all fruits, we can transfer the fruit semantics of apple and sentence to from the embedding for watermelon denoted as $e_{i}''$ by Eq.(3) in the main paper, so that the old realizes the transfer of information from old knowledge to new knowledge.
>
> **Q7**: Can you give definition of FWT and BWT?
>
> **A7**: Forward transfer (FWT)  is the influence of learning a task to the performance on the future tasks, while backward transfer (BWT)  is the influence of learning to the previous tasks. In this way, they can be formulated as: $FWT = \frac{1}{n-1} \sum_{i=2}^{n} h_{i-1,i}, BWT = \frac{1}{n-1} \sum_{i=1}^{n-1} (h_{n,i} - h_{i,i})$

---

> > ### Comment · Reviewer_pwUu · 2023-11-23
> >
> > Thank you for providing the additional intuitions here.

---

> ### Author Response · Authors · 2023-11-23
> **To  Reviewer pwUu**
>
> Regarding Q1, we appreciate the opportunity to clarify the different settings between previous work and CLKGE. We first give the definitions for temporal knowledge graph and dynamic knowledge graph.
>
> **Temporal knowledge graph (TKG).** A Temporal Knowledge Graph (TKG) is formally represented as $G_{TKG} = \{(v, r, u, t)\} \subseteq V_{TKG} \times R_{TKG} \times V_{TKG} \times T_{TKG} $, where $ V_{TKG} $ is the set of entities, $ R_{TKG} $ the set of relations, and $ T_{TKG} $ the set of associated timestamps. TKG completion tasks involve predicting the most probable entity $ u $ from $ V_{TKG} $ to complete a given query $ q = (v_{query}, r_{query}, ?, t_{query}) $.
>
> **Dynamic knowledge graph (DKG)**.  A Dynamic Knowledge Graph (DKG) is defined as $ G_{DKG} = \{(v, r, u)\} \subseteq V_{DKG} \times R_{DKG} \times V_{DKG} $, with $ V_{DKG} $ signifying the set of entities and $ R_{DKG} $ the set of relations. DKG queries are formulated to predict the entity $ u $ from $ V_{DKG} $ most likely to complete the query $ q = (v_{query}, r_{query}, ?) $. Moreover, DKGs evolve by adding or removing entities, relations, and triples, reflecting real-world dynamics.
>
> **Distinctions in Comparison Methods.** Regarding the comparison methods and CLKGE, there are two main differences in settings:
>
> 1. **Representation of Graphs**: Xu et al. and Jung et al. focus on TKGs represented as quadruples $(v, r, u, t)$, whereas CLKGE addresses DKGs, typically represented as triplets $(v, r, u)$. This fundamental difference in graph representation affects the experimental comparison.
>
> 2. **Entity Dynamics**: The methods by Xu et al. and Jung et al. do not accommodate the addition of new entities or the removal of old ones—a scenario that CLKGE explicitly handles. This distinction further complicates direct experimental performance comparisons.
>
> However, we have included a comprehensive discussion of these methodologies in the related work section of our revised manuscript to provide context and differentiate our unique contributions within the dynamic knowledge graph space.
>
> **References:**
>
> -  Xu W, Liu B, Peng M, et al. Pre-trained Language Model with Prompts for Temporal Knowledge Graph Completion[J]. arXiv preprint arXiv:2305.07912, 2023.
> -  Jung J, Jung J, Kang U. T-gap: Learning to walk across time for temporal knowledge graph completion[J]. arXiv preprint arXiv:2012.10595, 2020.
>
> We once again thank you for your insightful comments and suggestions.